# A bridge-like lipid transfer protein is critical for generation of invasive stages in malaria parasites

Andrés Guillén-Samander [1], Nika Perepelkina [1], Vendula Horáčková [1], Hannah M. Behrens [1], Hely O. Rodriguez Cruz [2], Joëlle Paolo Mesén-Ramírez [1], Ana Ribeiro-Holbein[1], Per Haberkant [3], Frank Stein [3] & Tobias Spielmann [1] ✉

Malaria blood stages build and maintain an intricate system of membranes during their cycle of rapid growth and schizogony (daughter-cell formation), requiring precise mechanisms of lipid synthesis and trafficking. Lipid transfer proteins (LTPs) at ER membrane contact sites (MCSs) have emerged as key for lipid distribution processes but remain largely unexplored in protozoans. Here we use the ER adapter VAP to identify essential mechanisms of lipid transfer at ER-MCSs in *P. falciparum*. One PfVAP-interacting LTP is the bridge-like PfVPS13L1, which allows bulk flow of lipids between two apposed membranes. PfVPS13L1 bridges the ER with the nascent inner membrane complex (IMC), a de novo-generated organelle required for schizogony. Its loss-of-function reduces IMC growth and leads to smaller anucleated progeny, impairing schizogony. Our data supports a model in which VPS13L1 is critical for the formation of apicomplexan invasive stages by mediating bulk transfer of lipids from the ER to the growing IMC.

*Plasmodium falciparum* parasites invade, grow and replicate in human red blood cells (RBCs), leading to exponential proliferation of the parasites in the blood and the symptoms of malaria. During their development in RBCs, the parasites establish and expand an intricate system of intra and extracellular membranes (Supplementary Fig. 1a), which requires precise mechanisms of lipid synthesis and distribution. In eukaryotic cells, lipids are mostly synthesized in the endoplasmic reticulum (ER) and are transported via vesicular trafficking or via lipid transfer proteins (LTPs)[1]. The latter happens primarily in regions of close apposition between organelles, so called membrane contact sites (MCSs), where LTPs are capable of extracting lipids from a bilayer into their hydrophobic cavity and deliver them to the apposed bilayer, thus enabling non-vesicle mediated transfer of lipids between organelles[2,3]. Eukaryotic LTPs can be classified as shuttle-like – proteins with a small hydrophobic cavity that can transfer 1-2 lipid molecules at a time usually engaging in counter transport reactions – or bridge-like –

proteins that span the entire distance between donor and acceptor membranes and have a large hydrophobic groove which accommodates many lipids, allowing for bulk transport of lipids to support membrane growth[1,2].

In recent years, cell biological studies in opisthokonts have revealed the importance of LTPs for various cellular processes involving membrane remodeling, homeostasis, biogenesis and repair[3,4]. Although similarity searches identified many LTPs in *P. falciparum* parasites[5], only two have been investigated: PfSTART1 and PfNCR1, both with sterol-transferring modules, and essential for the intraerythrocytic stage[6,7]. However, as both are transported beyond the plasma membrane (PM) of the parasite, the relevance of LTPs for intracellular membrane dynamics remains unknown. Importantly, PfSTART1 and PfNCR1 are also the target of novel antimalarials[8,9], highlighting the importance of studying LTPs for drug discovery. Additionally, MCSs have been observed between several organelles in *Plasmodium* and

[1]Bernhard Nocht Institute for Tropical Medicine, Hamburg, Germany. [2]Department of Cell Biology, Yale University School of Medicine, New Haven, CT, USA. [3]EMBL Proteomics Core, Heidelberg, Germany. ✉e-mail: spielmann@bnitm.de

other apicomplexans[10–12], but very little is known about potential lipid transferring processes, their relevance for the parasite and the proteins involved.

The ER, as a central hub of cellular lipid distribution, contacts and engages in lipid transferring processes with all other organelles in eukaryotic cells[3,13–16]. *P. falciparum* parasites are not expected to be an exception as MCSs between the ER and other organelles have been observed by different imaging techniques[12,17–19]. In plants and opisthokonts, the ER protein VAP plays an essential role in the formation of ER-MCS by acting as an ER adapter for proteins that contain a FFAT [two phenylalanines (FF) in an acidic tract (AT)] motif, many of which are LTPs[20–22]. In this study, we used proximity biotinylation to unbiasedly identify LTP partners of the *P. falciparum* ortholog of VAP (PfVAP). We identified a homolog of the bridge-like LTP (BLTP) VPS13, here named PfVPS13L1, that binds PfVAP via its N-terminal end and the inner membrane complex (IMC) through its C-terminal end, indicating localization to ER-IMC MCSs. The IMC is an organelle generated de novo during parasite schizogony, the process where up to 36 daughter cells (merozoites) are formed from a multinucleated cell. IMC biogenesis requires large amounts of lipids to wrap around the nuclei and delimit the newly formed merozoites, the stage invading new RBCs. Conditional inactivation of PfVPS13L1 impaired IMC growth and led to the generation of mini merozoite-like structures while a large part of the cytosol of the parent was left behind. Our work supports a model in which direct transfer of lipids in bulk from the ER to the growing IMC by the PfVPS13L1 bridge is essential to support IMC membrane extension and hence formation of functional progeny.

## Results

### PfVAP is an essential ER protein that binds FFAT motifs

Similar to opisthokont VAP, PfVAP (PF3D7_1439800) is a small protein with an N-terminal Major Sperm Protein (MSP) domain (responsible for FFAT motif binding in other organisms), followed by a coiled coil region and a C-terminal transmembrane domain expected to anchor the protein to the ER (Fig. 1a). Using the selection linked integration (SLI) system for genome modification, two parasite lines expressing a version of the protein fused to either Halo or a multipurpose GFP-tag (GFP-2xFKBP) on its N-terminus were generated (Halo-PfVAP^endo and N-GFP-2xFKBP-PfVAP^endo parasites, Supplementary Fig. 1b–c). Both cell lines showed a bright PfVAP signal in all stages of the asexual intraerythrocytic cycle of the parasite (see Supplementary Fig. 1a for relevant parasite organelles) and a clear localization to the ER, as shown by colocalization with the fluorescently tagged TM domain of Sec61β (mSc-TM^Sec61β) (Fig. 1b and Supplementary Fig. 1d), a protein commonly used as an ER marker in other organisms[18,23]. The ER marker surrounded the nuclei with additional features specific to different stages: in rings and trophozoites tubular extensions were observed as previously described[24], whereas schizonts contained several ER accumulations that disappeared after segmentation into daughter cells (Fig. 1b). By confocal microscopy, Halo-PfVAP largely colocalized with mSc-TM^Sec61β, but was enriched in some puncta along the ER (Supplementary Fig. 1e), a feature commonly reported for proteins that are engaged at MCSs[25–27]. More prominent hotspots were observed with GFP-2xFKBP-PfVAP (Supplementary Fig. 1d, f), resembling a phenotype observed in mammalian cells when ER proteins are tagged with GFP due to this tag's dimerization properties[28]. Despite these artifactual accumulations being seemingly innocuous for the parasites, we opted for using parasites with non-GFP tagged PfVAP for further analysis of localization and interactions.

During genome editing, loxP sites were added at both ends of the Pf*vap* gene (Supplementary Fig. 1b–c) to allow for gene excision with a rapalog-dimerization inducible Cre[29–32]. Growing synchronous GFP-2xFKBP-PfVAP^endo parasites with rapalog for a full cycle induced excision of the gene and led to loss of detectable PfVAP in ~80% of the parasites in the following cycle (Fig. 1c and Supplementary Fig. 1g). This

led to a ~90% reduction in growth over two cycles when compared to control (Fig. 1d), demonstrating the importance of PfVAP for intraerythrocytic cycle progression. The remaining growth may be due to the failure of full PfVAP removal in a percentage of parasites.

As in other organisms VAP acts as an ER receptor for FFAT-motif containing proteins[20], we tested whether FFAT binding is a conserved function of PfVAP. For this, we designed an assay in which the expression of a FFAT motif anchored to the PM (via a Lyn targeting sequence) would mediate the formation of ER-PM MCSs if it bound PfVAP (Fig. 1e). Indeed, when a conventional FFAT motif[20] was used in the assay, Halo-PfVAP and the ER in general were recruited to the PM in a section where the Lyn-EGFP-FFAT construct was enriched, but these features were not observed when a FFAT motif with the core phenylalanines mutated to alanines was used (Fig. 1f–g). These results indicate that the FFAT-motif binding capacity of the MSP domain of PfVAP is conserved and, more importantly, PfVAP was confirmed as a good candidate to identify FFAT-motif containing, potential lipid transfer proteins, in *P. falciparum*.

### Identification of VAP-binding proteins

In order to identify VAP interactors in the parasite, we carried out DiQ-BioID[33,34]. For this, we generated a cell line endogenously expressing PfVAP fused to two copies of FKBP (Supplementary Fig. 1b–c) and co-expressing the promiscuous biotinilysing enzyme miniTurbo[35], fused to mCherry and FRB for rapalog-mediated conditional dimerization to recruit it to PfVAP (Fig. 2a). This enables proximity labeling in an induction-dependent manner to subtract background labeling.

Quantitative MS analysis of biotinylated proteins on target (+rapalog) over control revealed a total of 185 significant hits in proximity to PfVAP (Fig. 2b–c, Supplementary Data 1). Given the high expression of this protein and its localization throughout the ER in all stages of the parasite, the detection of many hits is expected as this system identifies all proteins in proximity and not only direct interactors of PfVAP. Upon classification based on previous literature and protein function, the majority of proteins with a known or expected localization turned out to be ER proteins (Fig. 2c and Supplementary Fig. 2a–b), followed by other ER proximal compartments, such as the inner nuclear membrane (INM; the same compartment as the ER but facing the nucleoplasm, where VAP is also found in other eukaryotes[36]), the nuclear envelope embedded centriolar plaque (CP)[37], or the Golgi apparatus. Additionally, there was an overrepresentation of transmembrane proteins, expected due to the recruitment of the biotinyliser to the ER membrane (Fig. 2d and Supplementary Fig. 2c–f).

We modified the endogenous loci of 6 hits by adding a C-terminal Halo-SW tag (Halo-sandwich, corresponding to Halo flanked on each side by 2xFKBP) (Supplementary Fig. 2g), 4 of which were proteins of unknown localization and 2 with expected localizations[38,39]. Three of them showed a typical ER pattern, and three showed puncta in the surroundings of the nuclei (Fig. 2e), one of which was at the CP as shown by being adjacent to the inner CP marked by tubulin (Supplementary Fig. 2h), in agreement with a previous study in *P. berghei*[39]. All the localizations were compatible with a proximity to the ER and, hence, to PfVAP.

Additionally, we observed an overrepresentation of high-scoring FFAT motifs in the DiQ-BioID hits, likely indicating the identification of direct PfVAP binders (Supplementary Fig. 2i–j, Supplementary Data 2–3). We filtered the list of 185 hits based on their highest scoring FFAT motif sequences[40] and predicted AlphaFold3[41] interaction to identify 13 proteins containing motifs with a high likelihood of PfVAP binding, 9 of which were conserved in *Plasmodium spp* (Fig. 3a–b, Supplementary Fig. 2k–l and Supplementary Data 3).

To validate this strategy to find PfVAP binders, we used the above described PfVAP-binding assay (Fig. 1e–f) with the selected FFAT motifs. Out of 11 tested motifs, 7 resulted in the artificial formation of ER-PM MCSs (Fig. 3b–c), indicative of binding to PfVAP and a direct

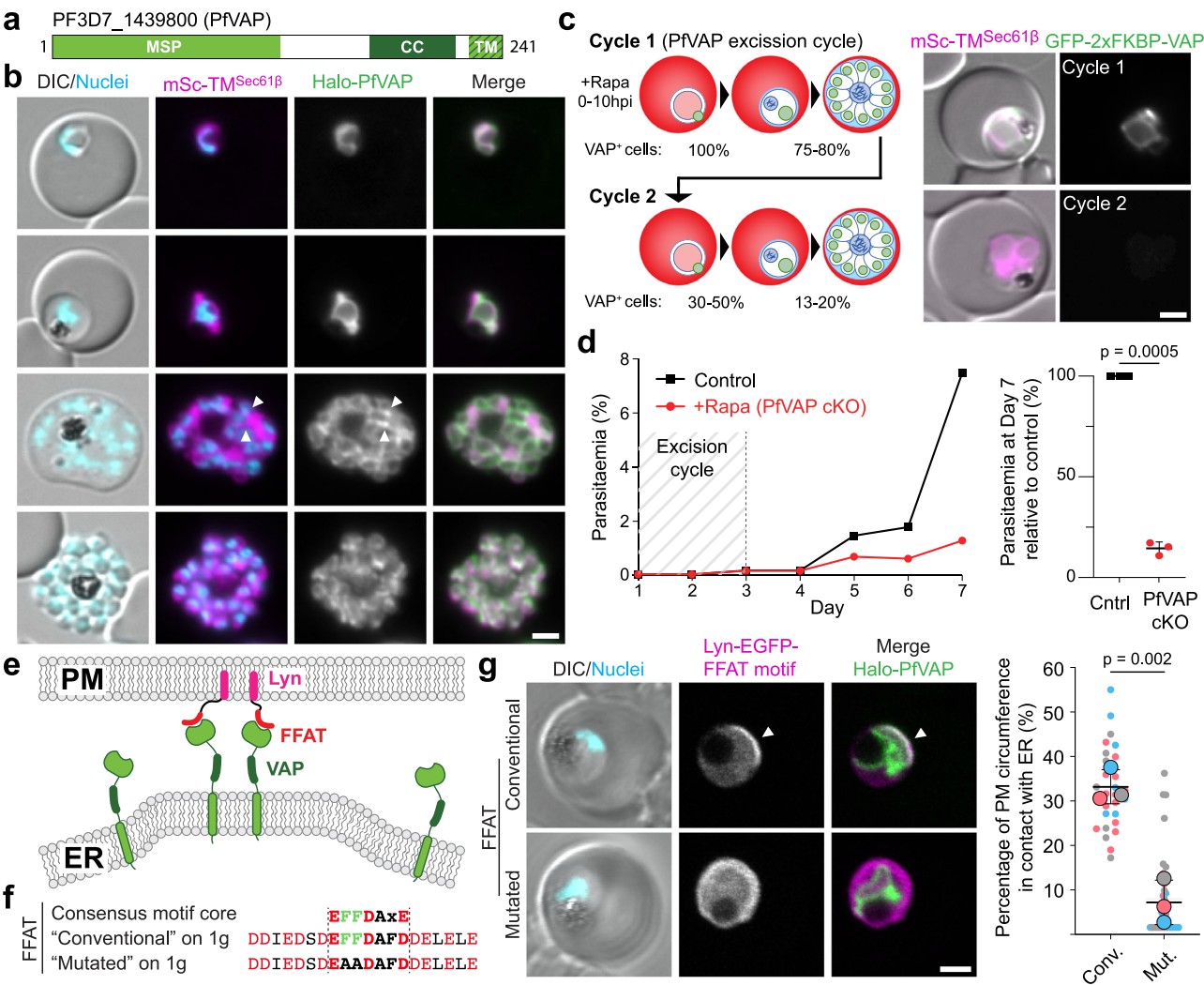

**Fig. 1 | PfVAP is an essential ER protein and binds FFAT motifs. a** Domain cartoon of PfVAP. MSP, Major Sperm Protein domain; CC, coiled coil; TM, transmembrane. **b** Fluorescence microscopy images of Halo-PfVAP[endo] parasites (Supplementary Fig. 1b–c) episomally co-expressing the TM domain of Sec61β fused to mScarlet (mSc-TM[Sec61β]). Arrowheads, PfVAP enriched in hotspots at ER (also in Supplementary Fig. 1e). **c** Schematic and quantification of PfVAP loss in GFP-2xFKBP-PfVAP[endo] parasites episomally co-expressing mSc-TM[Sec61β] after diCre-mediated gene excision induced using rapalog (+Rapa, PfVAP cKO) [left panel, 3-4 experimental replicates, with 36, 17, 62 and 38 parasites scored for the first timepoint (cycle 1, day 2); 53, 20, 68 parasites for the second timepoint (cycle 2, day 3); 61, 33, 41, 88 parasites for the third timepoint (cycle 2, day 4)] and representative fluorescence microscopy images from both cycles (right panel). **d** Growth of GFP-2xFKBP-PfVAP[endo] parasites with rapalog (PfVAPcKO) or without (control), measured by flow cytometry over 7 days (left panel) and growth relative to control

(Cntrl) on day 7 in 3 experimental replicates (right panel; bars, average and SD; p-value, two-tailed paired *t* test). **e** Schematic of the experimental design to test PfVAP and FFAT motif binding. The motif is anchored to the PM via Lyn, promoting ER-PM MCS formation if bound to PfVAP. **f** Conventional and mutated FFAT sequences used, based on the opisthokont consensus. **g** Representative confocal microscopy images of Halo-PfVAP[endo] parasites episomally expressing the indicated FFAT motif (left panel) and quantification of ER-PM MCSs distance (percentage of total PM circumference, right panel), shown as superplot[103], from *n* = 3 independent experiments (right panel) with a total of 28 and 29 parasites (1- or 2-nuclei trophozoite stage) quantified for the conventional (Conv.) and mutated (Mut.) motif, respectively; colors indicate independent experiments (small dots, individual parasites; large dots, average of each experiment; black lines, mean and SD; p-value, two-tailed unpaired *t* test of the means). DIC, differential interference contrast; Nuclei, Hoechst 33342; scale bars, 2 µm.

interaction of the corresponding full-length candidates with PfVAP. Alignment of the interacting motifs and their orthologous sequences in 10 different *Plasmodium* species revealed a slight difference from the consensus motif[40] (Fig. 3d): the amino acid in position 1, proposed to favor a negatively charged residue, was a hydrophobic residue in 5 out of the 7 motifs identified in *Plasmodium* proteins and, instead, the amino acid in position −1 was always negatively charged. This apparent importance of position −1 above that of position 1 was also suggested by testing point mutations (Supplementary Fig. 3a–b) and might be considered during future FFAT motif scoring. Of interest to our studies, of the 7 proteins here confirmed to have a PfVAP-interacting motif, 2 contained domains characteristic of LTPs (PF3D7_1131800 and

PF3D7_1346400, here named PfOSBP and PfVPS13L1, respectively), prompting us to study them in more depth.

**Lipid transfer proteins identified in proximity to the ER**

Besides the 2 FFAT-motif containing LTPs, 5 others proteins with domains typical for LTPs, resulting in 7 ER proximal LTPs, were identified in our PfVAP DiQ-BioID (Fig. 4a). Of these, 3 were shuttle-like and 4 bridge-like LTPs (Fig. 4b–d). The shuttle-like proteins all had lipid transfer domains belonging to different families: a VAD1 Analog of a StART (VASt) domain, a Phosphatidylinositol Transfer Protein (PITP) domain and an OSBP-Related Domain (ORD)[1,42] (Supplementary Fig. 4a–b). We named them, accordingly, PfGRAMD1 (PF3D7_0803600) based on its homology to the human GRAMD1

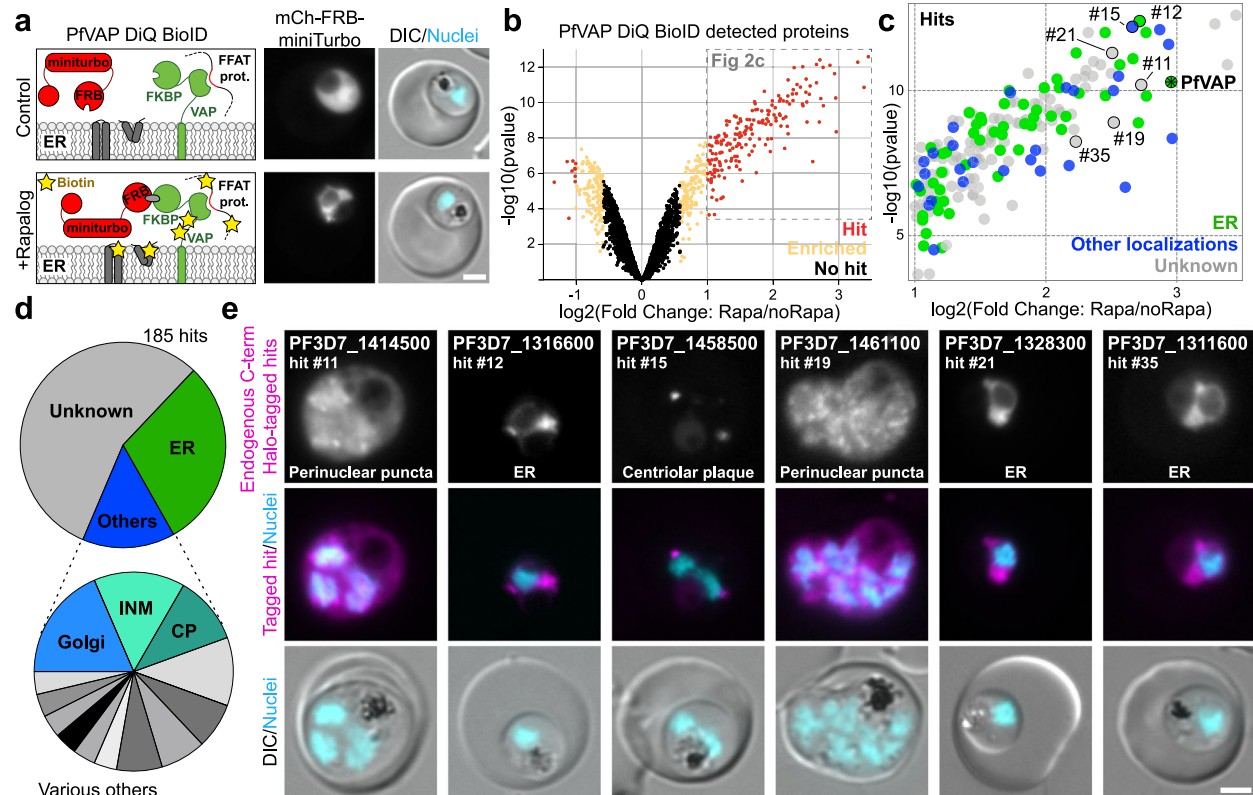

**Fig. 2 | PfVAP DiQ-BioID defines an ER-surface proteome. a** Schematic (left) and representative fluorescence microscopy images (right) showing the rapalog-mediated recruitment of the episomally expressed miniTurbo biotinilyzer to PfVAP for the identification of proximal proteins in 2xFKBP-PfVAP^endo parasites. **b** Volcano plot of the PfVAP DiQ-BioID outlined in (**a**) showing enrichment of proteins in plus rapalog (miniTurbo recruited to PfVAP) over control (miniTurbo free-floating in cytosol) color coded as indicated. Hits, absolute fold change of 2 or more and a false discovery rate (FDR) of 0.05 or less (185 proteins); enriched, absolute fold change >1.5, FDR <0.2 (252 proteins). Averages from *n* = 3 independent replicates analysed side by side in the same run (data in Supplementary Data 1). Moderated

two-tailed t-test was applied as implemented in the limma package. **c** Enlargement from (**b**) showing proteins considered hits color coded according to localization. Numbers indicate proteins analysed in (**e**). **d** Pie charts showing proportion of the hits in (**c**) with the indicated localization. Annotation of localization according to Supplementary Data 1B. INM: Inner Nuclear Membrane; CP: Centriolar Plaque. **e** Representative fluorescence microscopy images of parasites in which the indicated proteins (each corresponding to a hit numbered in (**c**)) were endogenously tagged with Halo-SW in their C-terminus. Classification of localization indicated. DIC, differential interference contrast; Nuclei, Hoechst 33342; scale bars, 2 μm.

proteins[43], PfPITP (PF3D7_1351000), which also has a second lipid transfer domain in its C-terminus with remote similarity to an ORD, and PfOSBP (PF3D7_1131800, containing a FFAT motif as shown in Fig. 3b–d).

We targeted the genes encoding the shuttle-like LTPs for endogenous tagging and obtained genome edited lines for PfPITP and PfOSBP, but not for PfGRAMD1, which we expressed episomally (Supplementary Fig. 4c). mCh-PfGRAMD1 and PfPITP-mNG localized to the ER, likely via their TM domains (Fig. 4e). Interestingly, both had other domains that can serve as lipid-binding adapters (PH and GRAM, Fig. 4c and Supplementary Fig. 4b), which could target the proteins to a specific ER-MCS in a regulated manner. Importantly, PfPITP also had a high-scoring FFAT motif but this did not bind PfVAP *in cellulo* (Supplementary Fig. 4d). PfOSBP-Halo-SW was mostly found in puncta proximal to the nucleus in trophozoites, colocalizing with the ER and sometimes overlapping with or adjacent to mNG-Rab6, a marker of the Golgi apparatus (Fig. 4e and Supplementary Fig. 4e–h). Interestingly, this localization was dynamic, as in some cells PfOSBP-Halo-SW was also found at the PM, while in other cells it was exclusively at the ER (Supplementary Fig. 4e–g) and this distribution changed throughout the parasite cycle (Supplementary Fig. 4f). Localization to the ER is likely mediated by its FFAT motif binding to PfVAP, and its additional PH domain could direct the protein to a specific ER-MCSs in a regulated manner (Supplementary Fig. 4b). Accordingly, episomal

expression of a tandem construct of its PH domain showed an enrichment at the Golgi, as seen by colocalization with a tagged adapter protein complex subunit AP-1μ (Supplementary Fig. 4i). In humans, the PH domain of OSBP binds Phosphatidylinositol 4-phosphate (PI4P) through a surface that is conserved in the PH domain of PfOSBP[44,45] (Supplementary Fig. 4j). Similarly to other eukaryotes, in *P. falciparum* parasites PI4P is enriched at the Golgi, and to a lesser extent at the PM[46]. It is therefore likely that PfOSBP is recruited dynamically to ER-Golgi MCSs or ER-PM MCSs depending on PI4P levels in these membranes.

*P. falciparum* proteome-wide HHpred searches revealed four bridge-like LTPs (BLTPs) homologous to the opisthokont VPS13 proteins (here named PfVPS13L1-4) of which PfVPS13L1 and PfVPS13L2 were present in the PfVAP DiQ-BioID (Fig. 4d, Supplementary Fig. 5a). These proteins had sequence homology in the characteristic N-terminal chorein domain. By using AlphaFold3[41] to predict the structures of all the large proteins of *P. falciparum* not included in the AlphaFold database (i.e., above 2700 aa) two more VPS13 like proteins (named PfVPS13L5 and L6) were identified (Supplementary Fig. 5a–b). Both of these were present in the PfVAP DiQ-BioID. In the case of PfVPS13L6, RBG repeats were observed throughout the structure, but these lacked the bridge-like organization normally seen for VPS13 proteins. This could be due to the extensive disordered regions in many *P. falciparum* proteins which complicate structural predictions.

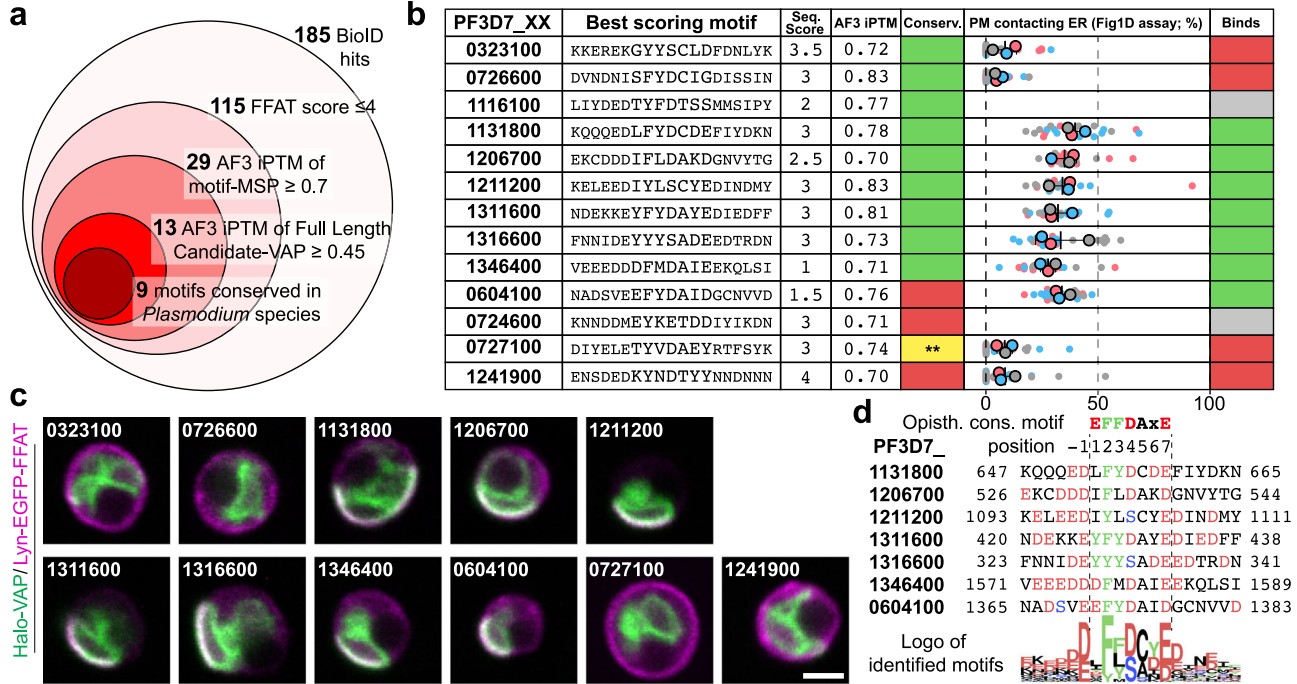

**Fig. 3 | PfVAP DiQ-BioID identifies FFAT-motif containing proteins. a** Schematic of hit filtering based on FFAT motifs starting with the 185 hits of the PfVAP DiQ-BioID (Fig. 2a–c) as indicated in the materials and methods and Supplementary Data 3. Scoring cut-offs are indicated; iPTM, interface predicted template modeling. **b** Table summarizing the 13 highest scoring FFAT motifs from (f) (Seq. Score, FFAT score based on Slee, J.A. & Levine, T.P[40]. AF3 iPTM, MSP domain of PfVAP AlphaFold3 iPTM value[41]) and experimental data shown in (**c**) quantified and displayed as a superplot[103] as in Fig. 1g [3 independent replicates and a total of at least 20 parasites (1- or 2-nuclei trophozoite stage, exact number indicated below next to *p*-value); colors indicate independent experiments (small dots, individual parasites; large dots, average of each experiment; black lines, mean and SD); binding was considered when there was no significant difference ($p \geq 0.05$) with the positive control (Fig. 1g) and indicated in green (gray, not tested; red, no binding), two-tailed unpaired *t* test of the means (exact *n* and *p*-values: PF3D7_0321300, 25,

0.0026; PF3D7_0726600, 31, 0.0003; PF3D7_1131800, 23, 0.0917; PF3D7_1206700, 20, 0.5591; PF3D7_1211200, 27, 0.727; PF3D7_1311600, 25, 0.8825; PF3D7_1316600, 25, 0.9431; PF3D7_1346400, 25, 0.1544; PF3D7_0604100, 27, 0.7238; PF3D7_0727100, 24, 0.0012; PF3D7_1241900, 25, 0.0015)]. Conservation of motifs (Conserv.) is indicated in green (no conservation in red); ** (yellow), motif that is conserved but part of a folded region, hence unlikely to bind PfVAP.
**c** Representative confocal microscopy example images of Halo-PfVAP^endo parasites episomally expressing the FFAT motifs indicated and used to quantify the resulting ER-PM MCSs distance as in Fig. 1g and summarized in (**b**). **d** Alignment of the 7 binding motifs identified and a sequence logo built from their conserved homologous sequences in 10 different *Plasmodium* species. Amino acids are colored according to their properties: green, aromatic; red, acidic; blue, phosphorylatable. Scale bar, 2 μm.

For this reason, we used AF3 to predict the structure of the VPS13L6 orthologue in *P. gallinaceum* which revealed the presence of RBGs folded as a VPS13-typical bridge (Supplementary Fig. 5c–d). We conclude that there is a total of six VPS13-like proteins in *P. falciparum*, which are conserved in other *Plasmodium* species and we have named PfVPS13L1-6 (Supplementary Fig. 6a).

We here focused on the VPS13-like proteins in the PfVAP hit list (PfVPS13L1, 2, 5 and 6). Unfortunately, we could not gather data for PfVPS13L5 as the SLI-mediated tagging did not yield any parasites. PfVPS13L2-Halo-SW signal was undetectable and PfVPS13L6-Halo-SW showed a very low fluorescence signal suggesting a cytosolic localization with accumulations (Fig. 4e and Supplementary Fig. 6b). PfVPS13L2 also had a high scoring FFAT motif but this did not bind PfVAP *in cellulo* (Supplementary Fig. 4d). Parasites where PfVPS13L1 was tagged with a Halo-SW (from here on referred to as PfVPS13L1-Halo-SW^endo, and PfVPS13L1-GFP-SW^endo for the GFP version) also gave a low signal but in early schizont stages, when its expression peaked, a punctate localization proximal to the nuclei was clearly discernable (Fig. 4e and Supplementary Fig. 6b). Since PfVPS13L1 has a functional FFAT motif (Fig. 3b–d), it is expected to be bound to PfVAP, and the punctate pattern could reflect its enrichment at an ER-MCSs with another organelle.

We then determined the essentiality of the two FFAT-motif containing LTPs PfOSBP and PfVPS13L1 that localized in puncta that may represent ER-MCSs. For this, we episomally expressed a Lyn-fused FRB

construct in the respective lines. As the SW constructs have four copies of FKBP, addition of rapalog promotes the recruitment of the LTPs to the PM, allowing us to study their loss-of-function by mislocalization (Knock-sideways, KS[32,47,48]). Mislocalization of the target protein to the PM was efficient in both cell lines (Supplementary Fig. 6d–e). This had no effect on parasite growth in the case of PfOSBP (Fig. 4f and Supplementary Fig. 6c–d), in accordance with a previous study where a targeted gene disruption was achieved[49]. In contrast, removal of PfVPS13L1 was lethal for the parasites (Fig. 4f and Supplementary Fig. 6c, e), which prompted us to focus our further investigation on this BLTP's function. Knock-sideways was also done for the non-FFAT containing PfVPS13L6, but this protein turned out to be dispensable (Supplementary Fig. 6c, f).

## PfVPS13L1 bridges the ER and the IMC
Similar to other VPS13 proteins[4], PfVPS13L1 is a very large protein (5988 residues) predicted by AlphaFold3[41] to fold as a rod formed by a twisted β-sheet which features three folded domains on its C-terminal end: (1) a VPS13 adapter binding (VAB) domain that sticks out of the rod and is formed by six β-sheet repeats, (2) an ATG2-C, formed by amphipathic helices, and (3) a PH domain; the latter two domains arranged in tandem at the C-terminal end of the rod. In addition, PfVPS13L1 is predicted to have a UBA domain on the N-terminal end of the VAB domain, similar to metazoan VPS13D proteins[50,51] (Fig. 5a–c, Supplementary Fig. 7a–e and Supplementary Movie 1). The ~22 nm rod

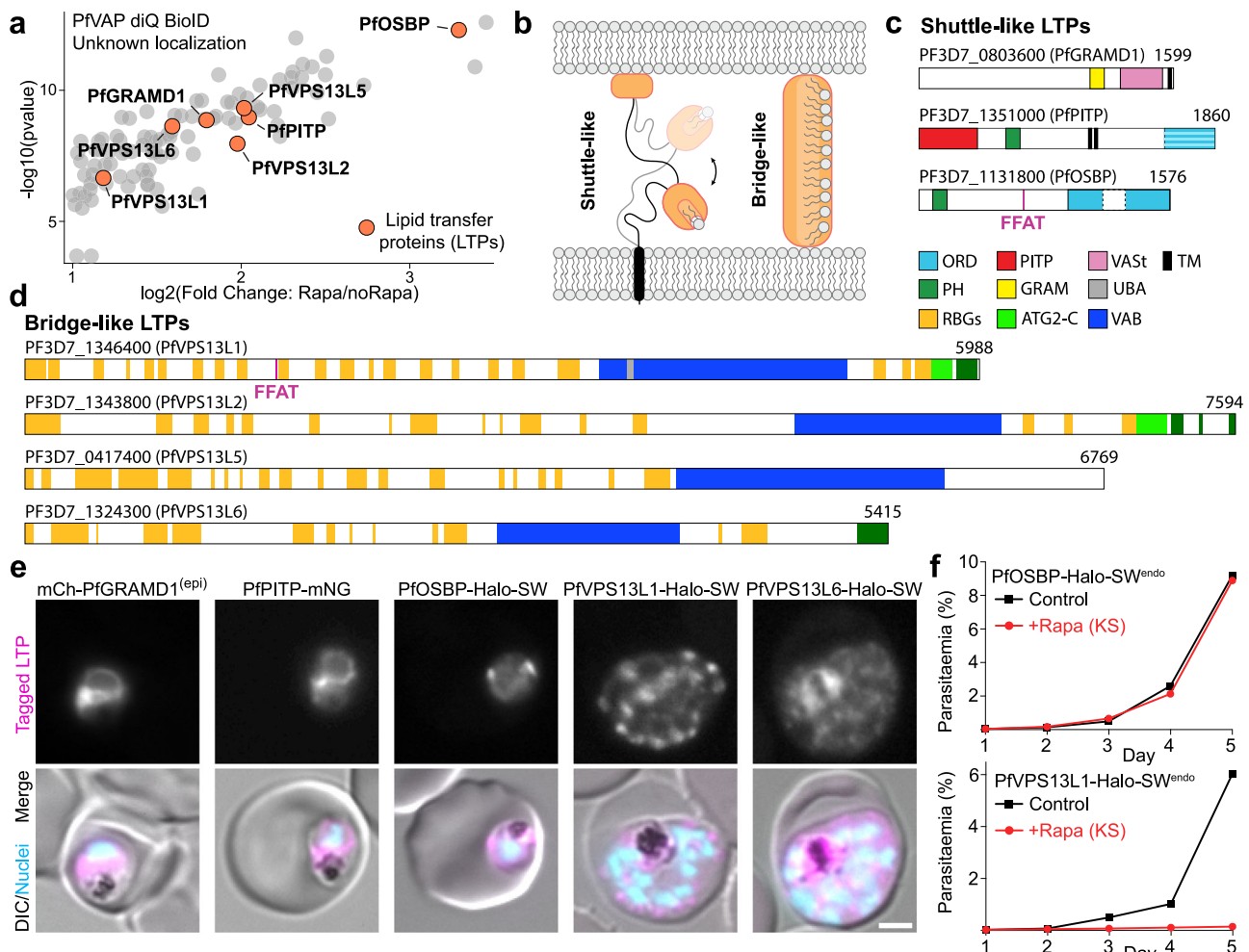

**Fig. 4 | Lipid transfer proteins (LTPs) in proximity to PfVAP. a** 7 LTPs identified in the PfVAP DiQ-BioID unknown hits (plot as in Fig. 2c). **b** General sketch of shuttle- or bridge-like LTPs. **c–d** To scale representation of the 3 shuttle-like (**c**) and 4 bridge-like LTPs (**d**; all remote homologs of opisthokont VPS13) highlighted in (**a**) showing the domains identified in the AF3 predicted structures. TM, transmembrane domain. Numbers, total amino acid sequence length. In (**d**), β-strands presumably forming the lipid transfer groove are in orange, even if a full RBG domain was not observed (more detailed RBG domains annotation of PfVPS13L1 in Fig. 5a and

Supplementary Fig 7a). **e** Representative fluorescence microscopy images of parasites episomally (epi) (mCh-PfGRAMD1) or endogenously expressing the indicated tagged LTPs. **f** Growth of parasites after knock-sideways (KS, rapalog) of the indicated PfVAP binding LTPs compared to control, measured by flow cytometry (*n* = 3 (PfVPS13L1) or 4 (PfOSBP) independent experiments; Comparison of day 5 parasitemia for all replicates in Supplementary Fig. 6c). DIC, differential interference contrast; Nuclei, Hoechst 33342; scale bars, 2 μm.

is formed by a series of 12 repeating β-groove (RBG) domains, a repeating unit of 5 antiparallel β-strands and 1 α-helix (except for the first and last RBG domains which have fewer β-strands) (Supplementary Fig. 7a–b), which gives the name to the superfamily VPS13 belongs to [52,53]. Although the size of its predicted 3D structure is similar to that of yeast Vps13 or human VPS13A (both also with 12 RBG domains), PfVPS13L1 comprises around twice the number of amino acids[4], largely due to flexible stretches protruding from the folded core (compare Fig. 5b–c). This, in addition to the further divergence of the other VPS13L protein sequences, hampered the proper prediction of their full structures.

The RBG domains have a hydrophobic surface facing the internal side of the twisted β-sheet that forms the rod (Fig. 5d and Supplementary Fig. 7b), which in BLTPs of other organisms has been shown to be able to accommodate the hydrophobic tails of several lipids[54–56]. When the rod is located between two organellar membranes, lipids would be able to flow in bulk from one organelle to the other. The FFAT motif of PfVPS13L1 is found in a flexible stretch on the fifth RBG domain, localized closer to the N-terminal end of the rod (Fig. 5c and

Supplementary Fig. 7f), suggesting that this end is bound to the ER (Fig. 5e). Interestingly, 3 out of the 4 human VPS13 proteins have an FFAT motif found in a similar region (right before the 5th RBG domain), which also leads to the attachment of their N-terminal end to the ER, considered the lipid donor membrane[4]. On the C-terminal end of the protein, the specific interactions of the adapter domains (VAB, PH domains) with protein or lipid partners usually dictate the identity of the receiving membrane (Fig. 5e). The amphipathic helices of the ATG2-C C-terminal domain are thought to partially insert themselves into the bilayer, potentially facilitating the lipid delivery process[57].

VPS13 proteins are expected to mediate bulk flow of lipids and are essential for the de novo organelle biogenesis in other organisms[4]. In *P. falciparum*, the inner membrane complex (IMC) is generated de novo around the nuclei of schizonts, the stage where we observed the peak of PfVPS13L1 expression (Supplementary Fig. 8a). A previous BioID detecting IMC proteins in *P. falciparum* contained PfVPS13L1 as a top candidate[58], prompting us to assess its localization in relation to the IMC marker mNG-PhIL1[59]. PfVPS13L1-Halo-SW was enriched at foci which colocalized with a subsection of

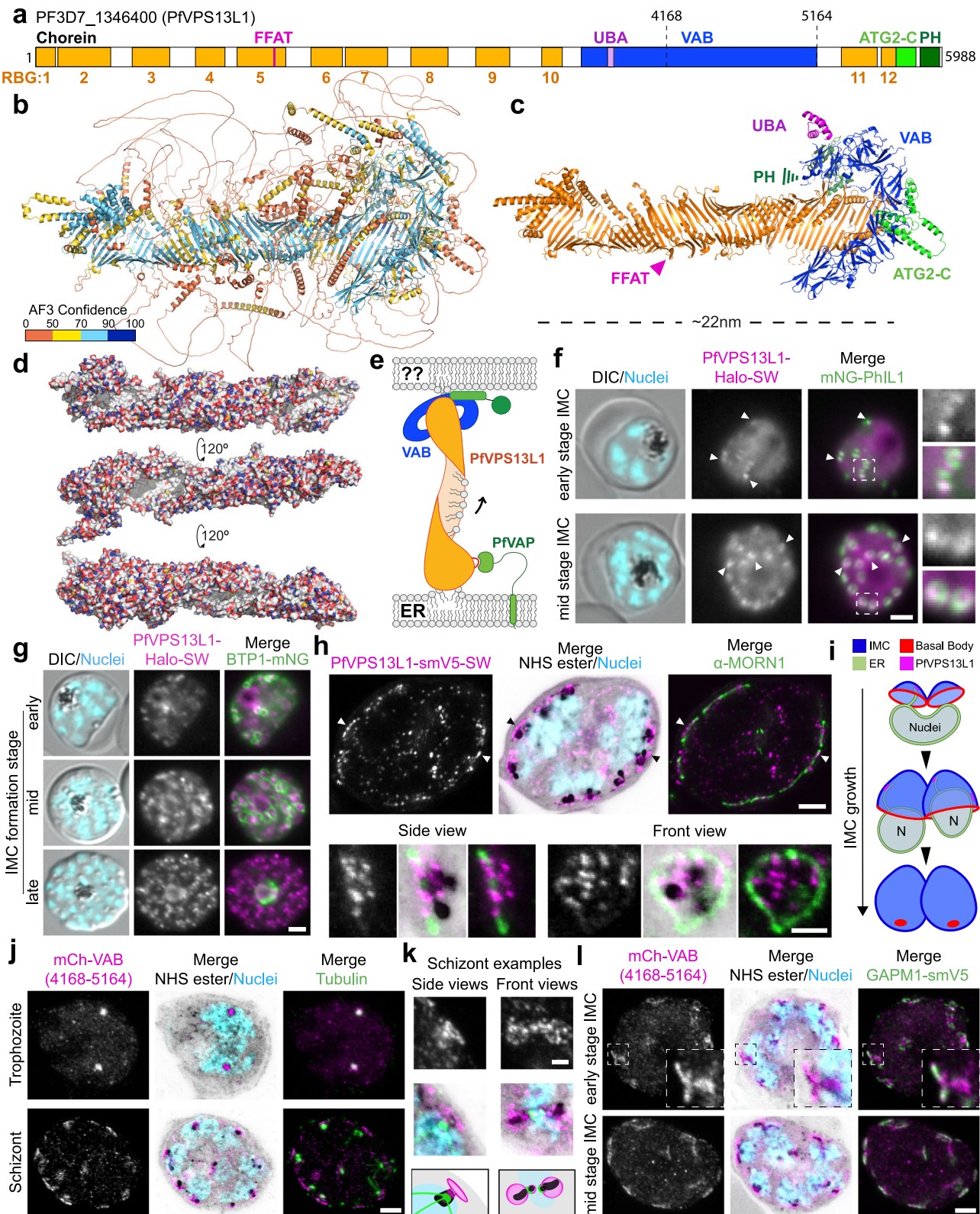

the IMC at early and mid stages of its formation (Fig. 5f and Supplementary Fig. 8a–c), which by confocal microscopy sometimes appeared to be on the IMC side facing the CP (Supplementary Fig. 8b), and were also directly colocalizing or adjacent to the ER (Supplementary Fig. 8d–e), compatible with the idea that PfVPS13L1 was present at the MCS between both organelles. These foci were in close proximity of BTP1-mNG, a marker of the basal complex (BC), which delimits the leading edge of the growing IMC[60], during early

stages of IMC formation when IMC and BC markers are not easily distinguishable (Fig. 5g). However, this proximity decreased as IMC growth progressed (Fig. 5g). Ultrastructure expansion microscopy (U-ExM) confirmed that smV5-tagged PfVPS13L1 (Supplementary Fig. 6b) foci were localized within the BC delimited area (as determined by the BC marker MORN1) but did not exclusively colocalize with it (Fig. 5h), confirming that PfVPS13L1 interactions with the IMC, and hence ER-IMC MCSs, were not in relation to the BC (Fig. 5i).

**Fig. 5 | PfVPS13L1 bridges the ER to the nascent IMC. a** Domain cartoon of PfVPS13L1, indicating the RBG domains that compose the lipid transferring rod, the FFAT motif and the C-terminal adapter and membrane binding domains. Black numbers, amino acids; orange numbers, RBG domains. **b** Ribbon representation of AlphaFold3 predicted structure of PfVPS13L1 colored by prediction confidence. **c** Structure from (**b**) with extensive flexible regions removed for clarity, and colored by domain annotation. **d** Surface representation of the rod composed by the 12 RBG domains colored by element (oxygen, red; nitrogen, blue; carbon, white), highlighting the internal white (hydrophobic) groove. **e** Schematic of PfVPS13L1 expected recruitment to an ER-MCSs based on its N-terminal FFAT motif binding to the ER via PfVAP. **f–g** Representative fluorescence microscopy images of parasites expressing PfVPS13L1-Halo-SW from the endogenous locus and episomally expressing the IMC marker PhIL1 (**f**) or the BC marker BTP1 (**g**) fused to mNeon-Green (mNG-PhIL1 or BTP1-mNG). Arrowheads, overlap during IMC formation. Boxed areas are enlarged 2.5x. **h** Ultrastructure Expansion Microscopy (U-ExM)

images of parasites in mid-stage of IMC formation expressing PfVPS13L1-smV5-SW from the endogenous locus co-immunostained for α-MORN1. **i** Schematic of IMC extension showing BC migration and sites of ER-IMC contact. The schematic was inspired by a previously published illustration[104]. **j–l** U-ExM images of parasites expressing a construct of the C-terminal VAB domain of PfVPS13L1 (see Supplementary Fig. 7c) fused to mCherry (mCh-VAB) co-immunostained for α-tubulin (**j**, **k**) or episomally co-expressing GAPM1-smV5 (**l**). mCh-VAB colocalized with the centriolar plaque (CP, marked by tubulin and NHS density) in trophozoites but formed a ring-like structure adjacent to the outer CP in schizonts (**j**, examples enlarged in **k**) that colocalized with GAPM1-smV5 (**l**). The enlargements in k show examples where the ring was captured from side and front perspectives, with a schematic below simplifying the observation. U-ExM images are maximum intensity projections of 4 (**h**-top), 6 (**h**-bottom), 10 (**j**-top), 8 (**j**-bottom), 3 (**k**), 12 (**l**-top), and 6 (**l**-bottom) slices. DIC, differential interference contrast; Nuclei, Hoechst 33342; scale bars, 2 μm in [**f**, **g**, **h**-bottom & **k**]; 5 μm in [**h**-top, **j**&**l**].

As the C-terminal adapter domains of VPS13 proteins are expected to determine the organellar binding specificity, we sought to confirm the IMC interaction by episomally expressing constructs containing these domains of PfVPS13L1 (Supplementary Fig. 9a). The construct encoding two copies of the PH domain showed a cytosolic signal (Supplementary Fig. 9b), but the one containing the tandem of ATG2-C and PH domains showed an enrichment in small-rounded compartments also observable by DIC in late RBC stages, likely corresponding to lipid droplets (LDs) (Supplementary Fig. 9c). This is not surprising, as the amphipathic helices in the ATG2-C domain target LDs in other VPS13 proteins such as in human VPS13A and VPS13C[26], but this is not expected to be the main site of action for these proteins as their role in LD homeostasis is minimal[61]. The construct containing the VAB domain displayed a clear punctate localization adjacent to the inner centriolar plaque (CP) indicated by a tubulin marker in trophozoite and schizont stages, and dispersed at the end of parasite segmentation (Supplementary Fig. 9d), similar to what is known to occur with the outer CP[62]. U-ExM revealed that while this signal colocalized with the CP in trophozoites, it acquired a ring-like morphology in schizonts that was adjacent to the CP, on its apical side (Fig. 5j–k), where the nascent IMC generates from[60]. Accordingly, this signal appeared to localize between the CP and the Golgi, partially colocalizing with an IMC marker in confocal microscopy (Supplementary Fig. 9e–g), and by U-ExM the VAB ring was observed colocalizing with the early stages of the IMC, as marked by the IMC protein GAPM1 (Fig. 5l). A recent publication reported the existence of a ring-like IMC initiation scaffold near the apical side of developing merozoites, adjacent to the CP, that disperses in late schizont stages[63], which would be in accordance with what we observe for the VAB domain of PfVPS13L1.

To confirm the IMC connection of PfVPS13L1, we performed DiQ-BioID to identify proteins in proximity to the C-terminal end of the PfVPS13L1 bridge (Supplementary Fig. 10a–b, Supplementary Data 4). Indeed, our DiQ-BioID results showed an enrichment of bona fide IMC proteins and proteins expected to be in the IMC (based on previously published proximity biotinylation experiments[58]) (Fig. 6a and Supplementary Fig. 10c–d). One protein, PF3D7_1364000, was consistently the highest enriched hit, suggesting a potential interaction. This protein contains two pore-forming domains of the Aegerolysin family[64,65] and hence here was named PfAegerolysin.

Genome modified parasites expressing PfAegerolysin with a mNG-SW tag revealed that this protein localizes to the IMC, most prominently in a subsection (Fig. 6b and Supplementary Fig. 10e–f), similar, albeit brighter, to what was observed for PfVPS13L1. U-ExM revealed that similarly to PfVPS13L1, PfAegerolysin-smV5-SW (Supplementary Fig. 10e) was detected throughout the IMC (Fig. 6c–d, Supplementary Fig. 10g), with no evidence of a specific enrichment to the BC (Fig. 6c–d and Supplementary Fig. 10g).

PfAegerolysin contains a C-terminal HEPN-like domain (HEPN-L; Fig. 6e) which was predicted by AlphaFold3 to interact directly with the

PH domain of VPS13L1 (iPTM=0.84) through surfaces conserved across apicomplexan species (Fig. 6f and Supplementary Fig. 10h). We tested this interaction by overexpressing the HEPN-L domain of PfAegerolysin fused to a Lyn PM-targeting sequence in parasites expressing a construct of two copies of the PH domain of PfVPS13L1 in tandem. This construct, which when expressed by itself was cytosolic, was recruited to the PPM by the HEPN-L domain (Fig. 6g–h) and mutation of two conserved residues in the interaction surface abolished the recruitment (Fig. 6h and Supplementary Fig. 10h–k). This confirms the AlphaFold3 predicted interaction.

Taken together, our data shows that the conserved FFAT domain of PfVPS13L1 binds the ER (Fig. 3b–c) while the VAB and PH domains on the other end of the protein bind the IMC (Figs. 5j–l and 6), the latter through an interaction with PfAegerolysin (Fig. 6). This indicates that PfVPS13L1 forms a lipid transferring bridge at MCSs between the ER around the nucleus and the growing IMC.

## PfVPS13L1 is essential for IMC formation and proper segmentation

To understand the function of PfVPS3L1, we first assessed in what stage of the intraerythrocytic cycle it was essential by inducing its mislocalization with rapalog in different stages. Despite a small percentage of parasites dying in the trophozoite stage after mislocalization of PfVPS13L1, the largest defect was observed in later stages which failed to produce new progeny, evident from an absence of new rings in the next cycle (Fig. 7a and Supplementary Fig. 11a). To confirm this observation, we induced the loss of PfVPS13L1 function around 36hpi (i.e., right before the schizont stage and hence IMC formation) and evaluated how many new rings the schizonts were able to produce in the next cycle. Almost no new parasites were observed after mislocalization of PfVPS13L1, indicating the essentiality of PfVPS13L1 for late intraerythrocytic stages, which could include segmentation, egress or invasion of new RBC (Fig. 7b-c).

As we had determined the localization of PfVPS13L1 to be at ER-IMC MCSs, the most parsimonious explanation would be that this BLTP is transferring lipids in bulk to support the de novo IMC formation. We evaluated this hypothesis by monitoring IMC growth upon loss-of-function induction in late stages. Live-cell confocal microscopy showed that, although an initial IMC seed was observed in all cases, the further growth of this membrane was severely reduced when PfVPS13L1 was not functional, leading to a faulty egress upon RBC rupture that left behind a large residual body (Fig. 7d–e and Supplementary Movie 2).

To better understand the morphological defects in the schizonts where PfVPS13L1 was mislocalized, we arrested the parasites shortly before egress by using Compound 2[66]. While in the control cells the IMC membranes formed and distributed all around the daughter parasites, always surrounding a nucleus, severe defects were observed upon PfVPS13L1-Halo-SW mislocalization: the IMC appeared smaller

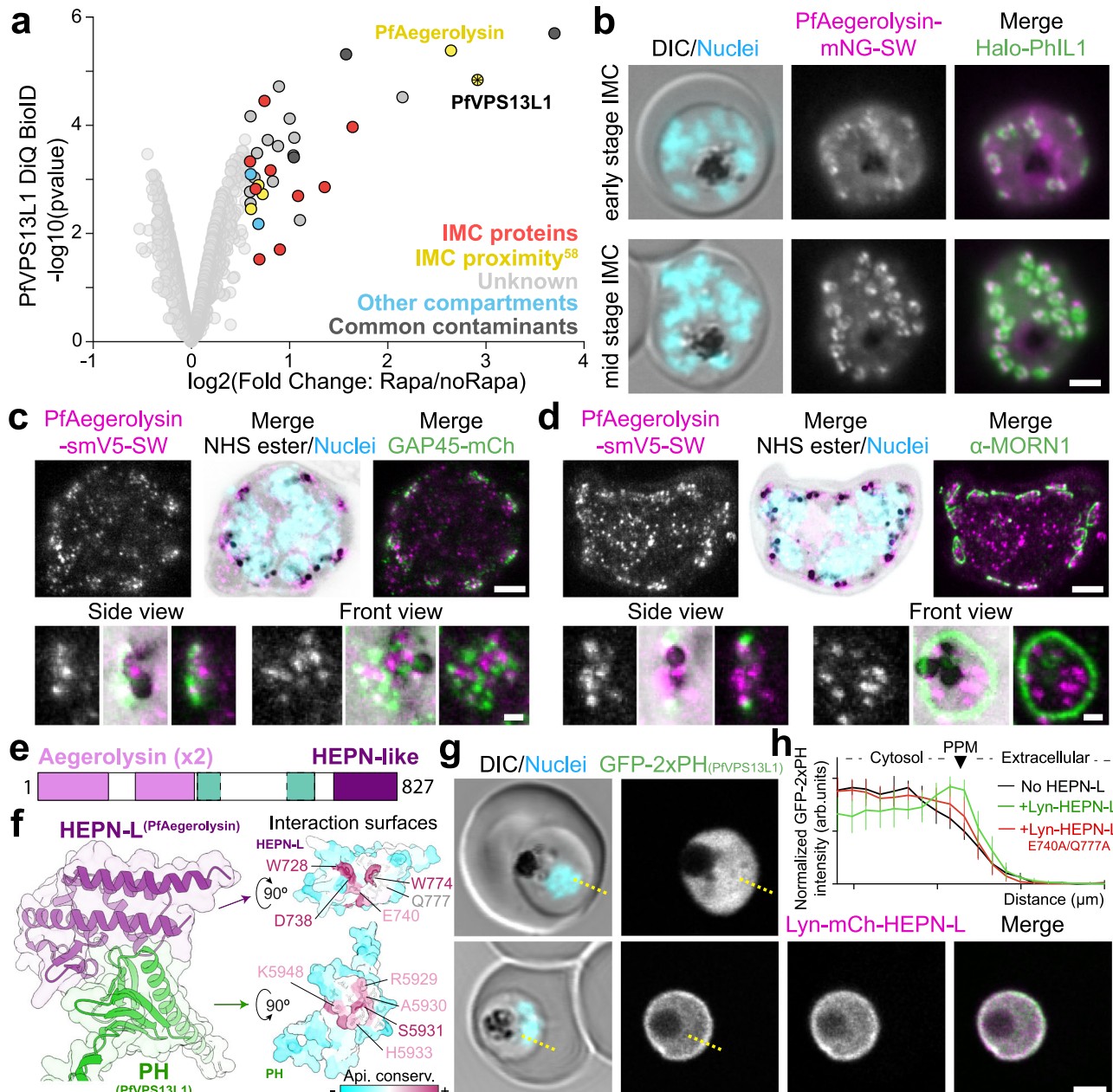

**Fig. 6 | C-terminal PH domain of PfVPS13L1 interacts with PfAegerolysin at the IMC. a** Volcano plot of DiQ-BioID of the C-terminus of PfVPS13L1 showing enrichment (3 independent samples per condition; second experimental repeat in Supplementary Fig. 10c–d) of proteins in plus rapalog over control as outlined in Supplementary Fig. 10a. Enriched proteins (fold change>1.5, *p* < 0.05) are highlighted with a black stroke and color-coded as indicated. Moderated two-tailed *t*-test was applied as implemented in the limma package. IMC proximity based on previous PhIL1 BioID[58]; common contaminants based on previous DiQ-BioIDs[33]. **b** Representative fluorescence microscopy images of parasites with endogenously mNG-SW-tagged PF3D7_1364000 and episomally expressing Halo-PhIL1, during IMC formation. **c–d** U-ExM images of parasites in mid-stages of IMC formation endogenously expressing PfAegerolysin-smV5-SW co-immunostained for GAP45-mCh (**c**) or α-MORN1 (**d**). **e** Domain cartoon of PfAegerolysin. Unknown domain in turquoise. **f** Alphafold3-predicted interaction between HEPN-L domain of PfAegerolysin and PH domain of PfVPS13L1 (iPTM=0.84). Left panel, interacting domains shown as ribbons. Right panel, domains separated and turned to show the interaction surface, colored by conservation across apicomplexan homologs, and the residues at the interaction interface shown as sticks. Conserved residues in the interface are highlighted. **g** Representative confocal images of parasites episomally expressing the PH domain of PfVPS13L1 (GFP-2xPH) with or without co-expression of the HEPN-L domain of PfAegerolysin artificially localized to the PM with a Lyn-targeting sequence. **h** Intensity measurements across a line transversing the PM, as in (**g**), of the GFP-2xPH construct when expressed by itself (black), or co-expressed with the Lyn-mCh-HEPN-L construct, either WT (green) or with two point mutations in the interacting surface (E740A/Q777A; red, representative images of the mutant HEPN-L in Supplementary Fig. 10i). Graph shows mean with SD for each pixel from *n* = 3 independent experiments for cells co-expressing the Lyn-mCh-HEPN-L constructs (WT=10 cells; mutant=14 cells) and *n* = 1 for expression of the PH domain alone (5 cells). U-ExM images are maximum intensity projections of 5 (**c**-top), 4 (**c**-bottom), 9 (**d**-top) and 2 (**d**-bottom) slices. DIC, differential interference contrast; Nuclei, Hoechst 33342; scale bars, 2 μm in (**b, g**); 5 μm in [**c**-top, **d**-top, **j, l**]; 1 μm in [**c**-bottom, **d**-bottom].

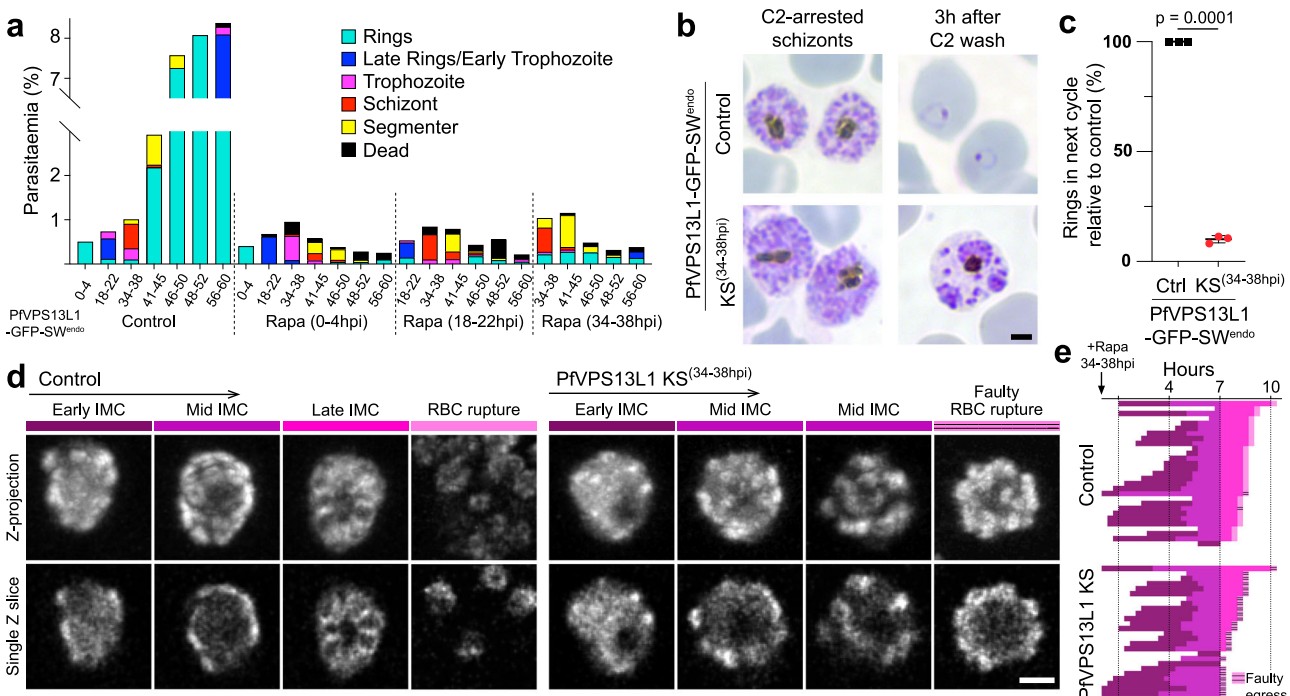

**Fig. 7 | PfVPS13L1 is essential during IMC formation. a** Parasite growth and stage progression assay in synchronous parasites (4 h stage window) upon PfVPS13L1 KS (using the PfVPS13L1-GFP-SW[endo] parasites (Supplementary Fig. 6b) episomally expressing the Lyn mislocaliser) induced at the indicated time points based on Giemsa smears (representative images are shown in Supplementary Fig. 11a). **b–c** Representative example Giemsa smears (**b**) and quantification (**c**) of rings produced after PfVPS13L1 KS (rapalog added at 34-38hpi) in comparison to no rapalog controls. Compound 2 was used to arrest segmenters in both conditions and rings assessed after wash out (*n* = 3 independent experiments; black lines, mean with SD; *p*-value, two-tailed paired *t* test). **d** Representative maximum intensity projections and single Z-slices of selected time points of 3D-timelapse imaging of PfVPS13L1-GFP-SW[endo] parasites episomally expressing Halo-PhIL1 (IMC) in control and PfVPS13L1 KS (+rapa) schizonts. PfVPS13L1 mislocalization triggered at 34-38hpi schizonts, imaging (20 min interval) started at 36-40hpi schizonts. **e** Quantification of time lapse experiment in (**d**). IMC development stages [indicated in (**d**)] were classified in each imaged schizont (*n*=29 and 27 cells, in control and KS, respectively, from 2 independent experiments) and progression represented graphically. Faulty egress: RBC rupture with merozoites not egressing or remaining attached to a large residual body. Scale bars, 2 μm.

and disconnected from the nuclei, with abundant DNA material and parasite cytoplasm being leftover in an atypically large residual body (Fig. 8a–b and Supplementary Fig. 11b). Segmentation still occurred but resulted in the formation of often anucleated small merozoite-like structures (Fig. 8a–c and Supplementary Fig. 11b–d). These defects ranged in severity, probably depending on the exact stage each individual schizont was in when PfVPS13L1 mislocalization was induced (Fig. 8a–d). U-ExM confirmed these phenotypes (Fig. 8e–f and Supplementary Fig. 11e–f). The U-ExM also showed that the rhoptries (another organelle formed in late stages) were still properly formed in the small defective merozoites and that tubulin appeared as extended structures characteristic of fully developed merozoites. These findings argue against a general development arrest and indicate a more exclusive effect of PfVPS13L1 loss of function in IMC formation (Fig. 8e–f and Supplementary Fig. 11e–f). Overall, this indicated a failure to sufficiently expand the IMC when PfVPS13L1 function was missing which in turn led to a failure to accommodate enough cytoplasm and a full nucleus into daughter cells during schizogony, leading to the generation of mini merozoite-like structures and leaving behind a large residual body (Fig. 9).

## Discussion

Protein-mediated direct lipid transfer between organelles is emerging as a key cell biological function in eukaryotes but has not been studied in malaria parasites—a protozoan of high medical importance with a complex life cycle involving rapid organelle expansion and extensive membrane rearrangements. Here we show that a VPS13-like BLTP is needed for the expansion of the IMC, a key structure underlying the PM of alveolates that in malaria parasite schizogony is generated de

novo and is needed as a scaffold for progeny formation and for host cell invasion[67]. Parasites without the function of this protein were incapable of generating functional progeny, suggesting lipid transfer fulfils key roles in the biology of this parasite and likely also in other apicomplexans. Our results support a model by which PfVPS13L1 enables the rapid and massive membrane extension of the IMC by supplying it with a bulk flow of lipids from the ER (Fig. 9). The N-terminal end of the bridge binds the ER via an interaction of its FFAT motif to PfVAP and the C-terminal VAB and PH domains bind the IMC (Fig. 9). In accordance with this model, MCSs between the ER and the IMC have been reported to occur in *P. falciparum* gametocytes, a stage where the IMC is largely expanded[12]. The disassociation between nuclei and IMC observed in the PfVPS13L1 loss-of-function phenotype would indicate that its lipid transfer function is coupled with a tethering role to maintain this MCS, as similar, albeit more extreme, disconnections have been observed upon the loss of proteins that hold the inner and outer centriolar plaques together[68].

Lipid transfer by VPS13 and other BLTPs is usually coordinated with lipid scrambling at the interacting membranes that ensure accurate distribution of lipids in the bilayers during their extraction and delivery[2,69,70]. While it remains to be determined which IMC TM protein could fulfil such a function, we identified proteins with multiple TM domains in the PfVPS13L1 C-terminal proxiome (Supplementary Data 4). The most enriched protein in the PfVPS13L C-terminal proxiome was PF3D7_1364000, a protein containing a pair of pore-forming aegerolysin domains[64,65], hence renamed here PfAegerolysin, which we found to directly interact with the PH domain of PfVPS13L1. Interaction between a BLTP and a pore-forming protein has to our knowledge not so far been reported and could suggest a functional coordination

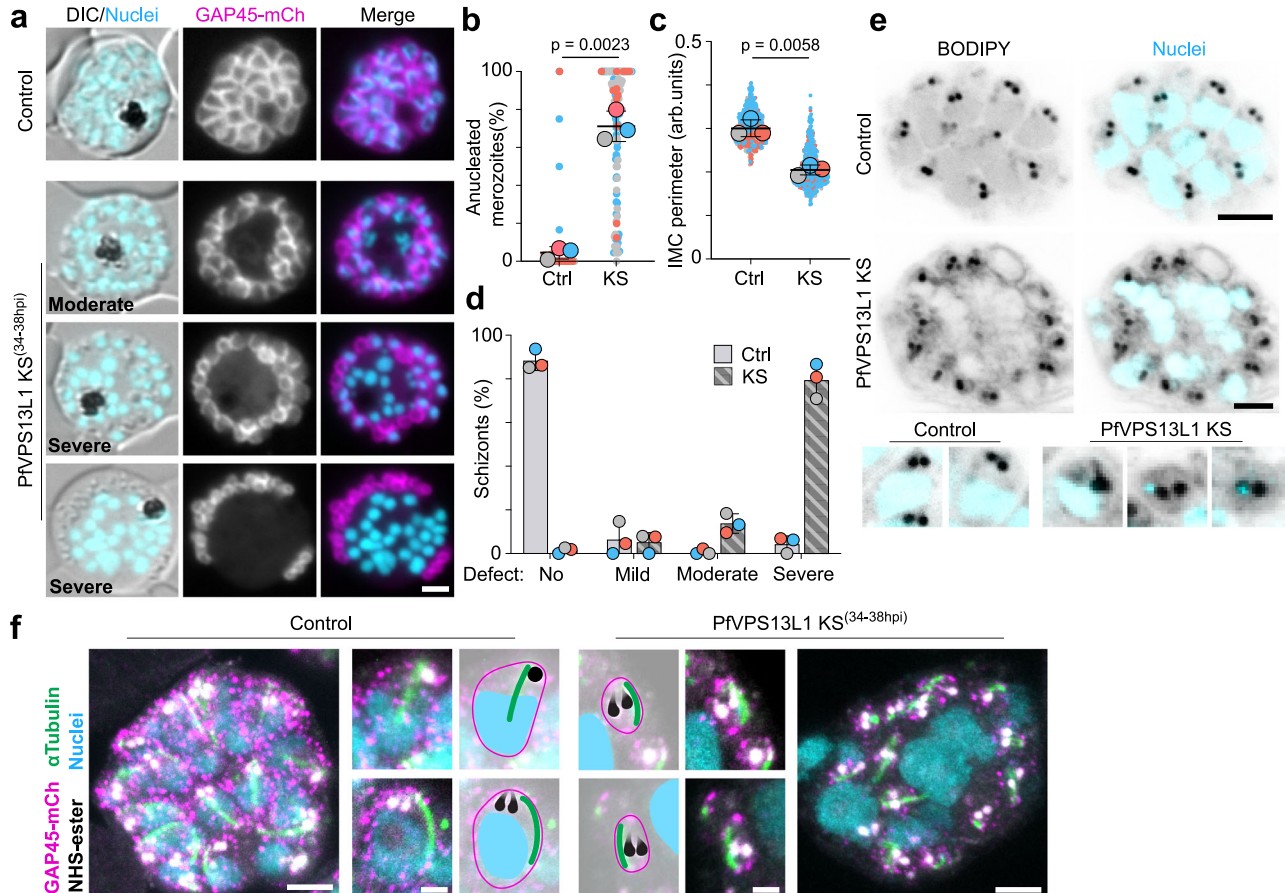

**Fig. 8 | PfVPS13L1 is essential for IMC extension and schizont segmentation.**
**a** Fluorescence microscopy images of control and late (34-38hpi)-induced PfVPS13L1 KS schizonts arrested before egress with compound 2. **b** Superplots[103] showing percentage of anucleated merozoites (defined as compartments delimited by the IMC marker without a nucleus) per schizont after KS of PfVPS13L1 compared to control (Ctrl). Three independent replicates with a total of 87 and 103 cells for control and KS, respectively; colors, independent experiments; individual parasites, small dots; average per experiment, large dot; black lines, mean and SD; p-values, two-tailed paired *t* test of the means. **c** Superplot as in (**b**) but measuring IMC perimeter. *p*-value, two-tailed paired *t* test of the means. All visible IMC compartments were measured in 15, 16, 16 control cells and 16, 19, 16 KS cells across three independent experiments. **d** Quantification of the phenotypes observed in (**a**) classified as mild (1-2 nuclei with no associated IMC), moderate (up to 6 nuclei with no associated IMC) or severe (at least half of the nuclei without associated IMC). Bar

graphs, percentage of each phenotype scored, (3 independent experiments, each represented by one color of dot); p-values (two-tailed paired t test of the means): 0.0016, 0.7134, 0.0055 and 0.0014 for no, mild, moderate and severe phenotype, respectively. **e**–**f** U-ExM images of control and late-induced KS schizonts stained with Bodipy (**e**) or NHS-ester, anti-mCh (for GAP45-mCh) and anti-αTubulin (**f**) showing accumulated nuclei in the residual body and anucleated merozoites upon PfVPS13L1 mislocalization (PfVPS13L1 KS). Some of the smaller merozoites carried a small amount of DNA (**e**, bottom panel). Enlargements show individual merozoites taken from different slices, including images with artificial graphical highlight of key features (rhoptries, black; IMC, magenta; Tubulin, green; nuclei, cyan). U-ExM images are maximum intensity projections of Z-slices, with 6 (**e**; Control and KS), 4 (**e**; enlargements), 13 (**f**; Control), 11 (**f**; KS) and 6 (**f**; enlargements) slices. DIC, differential interference contrast; Nuclei, Hoechst 33342; scale bars, 2 μm in [**a**, **f** (enlargements)] and 5 μm in [**e**, **f** (whole cell)].

between these membrane embedding domains and the bulk lipid delivery of PfVPS13L1 (Fig. 9). Interestingly, a human VPS13 homolog is recruited to the membranes of lysosomes upon rupture[71], indicating a tendency of these proteins to be recruited to porous membranes.

IMC and segmentation phenotypes have been observed upon the loss of function of well-established IMC proteins, such as the alveolins IMC1g and IMC1c, or the initial IMC scaffolding protein FBXO1, which upon removal lead to the formation of anucleated merozoites and large residual bodies[63,72]. Additionally, the localization of FBXO1 resembles that observed for the VAB domain of PfVPS13L1, which supports a lipid transfer function of PfVPS13L1 to expand the IMC from its early stages of formation. All of these studies point to a general role of the IMC in ensuring proper segmentation into progeny. Interestingly, impairing vesicular transfer disrupts the IMC in a different way, with IMC targeted proteins remaining cytosolic or localized to a different membrane, indicating the lack of an IMC membrane[73,74]. These observations do not exclude the role of the non-vesicular transfer

model of growth proposed here, because an initial IMC "seed", likely Golgi-derived[67], would be required for PfVPS13L1 to deliver lipids to (Fig. 9). In fact, membrane seeds are observed in similar organelle biogenesis processes. Namely, the growth of the autophagosome and of the prospore membrane in yeast, in which the BLTPs ATG2 and Vps13, respectively, transfer lipids from the ER to a Golgi-derived membrane seed[2,75–77]. The latter process is highly homologous to IMC formation, as the prospore membrane grows adjacently to the nucleus and the nuclear embedded CP, and in parallel to nuclear division during yeast meiosis[76]. These similarities, and this study as the first evidence for a BLTP function in organelle biogenesis in a protozoan, highlight this as a conserved mechanism for membrane expansion across all eukaryotes.

Although our phenotypic results are compatible with PfVPS13L1 fueling IMC growth through lipid transfer ER-IMC MCSs, some questions remain. Given the confined space in the parasite, particularly in schizonts, direct contact between the ER and IMC is difficult to

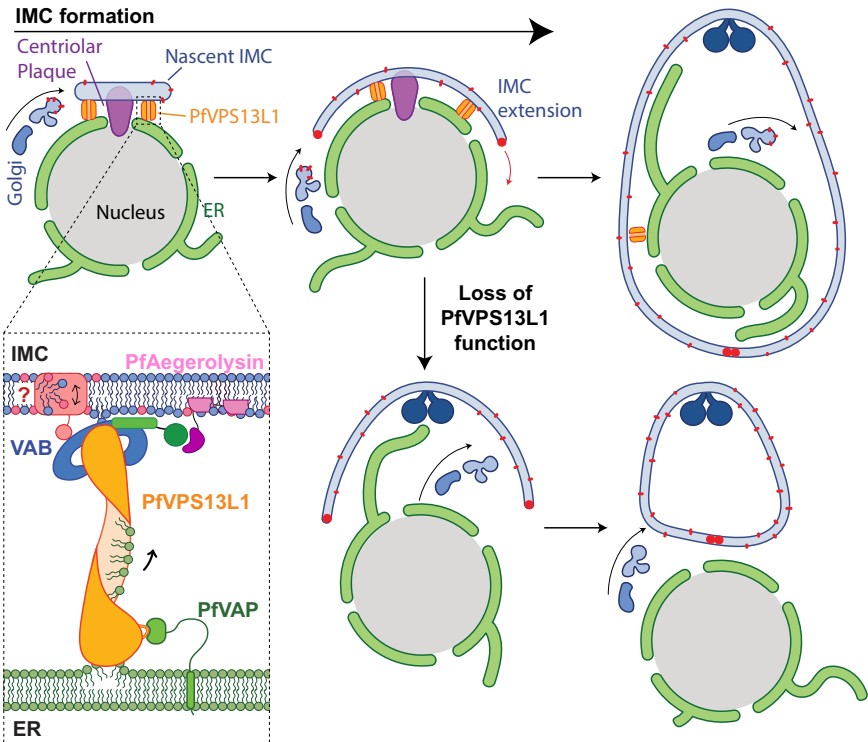

**Fig. 9 | Lipid transfer by PfVPS13L1 at ER-IMC membrane contact sites promotes IMC biogenesis.** Schematic summarizing findings of this study. PfVPS13L1 bridges the ER to the nascent IMC via binding of its N-terminal FFAT motif to PfVAP in the ER and of its VAB and PH domains to an unknown partner and PfAegerolysin, respectively, in the IMC generating a MCS and the direct transport of lipids to support IMC extension. Curved black arrows in the schematic indicates Golgi-derived vesicular trafficking and red arrow indicates migration of the basal complex.

determine and it is, for instance, unclear whether the contact is broken upon loss of PfVPS13L1. In addition, while knock-sideways allows for acute loss-of-function, it also involves the relocation of the BLTP to the PM, where a gain-of-function defect cannot be fully discarded. While episomal expression to complement the knock sideways would be an option to address this, doing this for such a large protein is challenging. Hence, the development of a genetic-based model for phenotypic rescue would be advantageous and could also open the possibility of studying the effect of specific mutations, such as e.g., in the interaction interfaces or on the hydrophilic bands across the RBG groove which produce a gain of function in yeast[4].

In addition to identifying PfVPS13L1, we also provide the proteome of the cytosolic face of the ER from living intraerythrocytic stages of *P. falciparum* parasites, obtained by proximity biotinylation using PfVAP as bait. These results showed conservation of FFAT-binding of VAP in the parasite and were used to refine the PfVAP-binding FFAT motif to present a genus-specific consensus. Seven proteins with an experimentally confirmed interacting motif were identified. Two of these proteins (PF3D7_1311600 and PF3D7_1316600) localized to the ER, even though they lack a TM region, and hence are expected to be peripherally recruited to the ER, congruent with a recruitment by PfVAP. Studies of other proteins with FFAT motifs could reveal previously unknown functions of the ER in intraerythrocytic stages. For example, one of the motif-containing proteins identified is PfSIP2 (PF3D7_0604100), an AP2 transcription factor binding heterochromatin known to be enriched at the inner nuclear membrane (INM)[78], where VAP has known interactors in other organisms[36]. Our study raises the possibility that PfVAP serves as a way to anchoring PfSIP2 and potentially chromatin to the INM.

Besides localizing PfVPS13L1 at the ER-IMC MCSs, our study also identified the LTP PfOSBP at the ER-Golgi MCSs, a function conserved for this protein in opisthokonts, and up to 5 other LTPs in proximity to the ER. This study thus largely expands the repertoire of studied *P.* *falciparum* LTPs and MCSs. As shown by previous studies[8,9], LTPs represent a potential target for antimalarial drugs that could act by blocking their lipid transfer capabilities.

## Methods

### Parasite culture, synchronization and transfection
*P. falciparum* 3D7 parasites were cultured in RPMI 1640 medium containing 0.5% Albumax II (Gibco #11021045), 10 mM Glucose, 12 mM sodium bicarbonate and 20 μg/ml Gentamycin (Ratiopharm), with a 5% haematocrit of human O+ erythrocytes at 37 °C and in a controlled gas environment of 5% O2, 5% CO2, 90% N2, following standard culture protocol. Transfusion human O+ blood was commercially purchased from Universitätsklinikum Hamburg-Eppendorf (Approval number 10569a/96-1). Age, sex and identity of blood donors was not known.

For synchronization of ring stage parasites, cultures were centrifuged at 2300 g for 3 min and the erythrocyte pellet was resuspended in 10 pellet volumes of 5% sorbitol, incubated for 10 min at 37 °C, followed by one wash with medium. For enrichment of late schizont stage parasites, 10 ml of cultures were slowly overlaid on 3 ml of 60% Percoll, and centrifuged at 2300 g for 3 min without brake. The layer of schizonts above the Percoll was collected and washed with medium once.

For transfections, Percoll-enriched late schizonts (originating from 400 μl of erythrocytes at 5% parasitaemia per transfection) were resuspended in 90 μl of a buffer containing 90 mM NaPO4, 5 mM KCl, 0.15 mM CaCl2 and 50 mM HEPES pH 7.6, and electroporated with 50 μg of plasmid in 10μl of TE buffer, using the Amaxa system (Lonza Nucleofector II AAD-1001N, program U-033) as previously described[79]. Electroporated parasites were incubated shaking (800 rpm) at 37 °C with 200 μl of fresh erythrocytes and 500 μl of medium for 30–60 min and were then transferred to standard culture conditions. A day after transfection, parasites containing the plasmid were selected with the corresponding drug (4 nM WR99210 (Jacobus Pharmaceuticals),

0.9 µM DSM1 (Merck #5.33304.0001), 400 µg/mL G418 (Merck #A1720), or 2 µg/µl blasticidin S (Invitrogen #R21001) for plasmids encoding the hDHFR, yDHODH, Neo-R, or BSD resistance genes, respectively). Medium and drugs were changed daily for 4 days, and parasites were periodically monitored.

## Plasmids

Plasmids generated for this study were cloned by Gibson ligation reaction (a mixture of T5 exonuclease (Epicenter #T5E4111K), Phusion DNA polymerase (NEB #M0530S), Taq DNA ligase (NEB #M0208L), dNTPs (Roth #K039.1), NAD (Sigma-Aldrich #n6522), DTT, MgCl$_2$ and Tris-HCl pH 7.5), linearizing a backbone with restriction enzymes and ligating in the insert that was PCR amplified with primers containing at least 20 nucleotides of overlap with the target backbone. In cases where the insert was smaller than 80 nucleotides, these were ordered to be synthesized as oligonucleotides including the overlapping regions. All oligonucleotides used were ordered from Sigma-Aldrich and all restriction enzymes used were ordered from NEB. In cases where a re-codonized sequence was required, synthesized DNA fragments were ordered from BioCat GmbH (Heidelberg, DE). A list of all the plasmids generated in this study and their respective cloning reactions is found in Supplementary Data 5, and the expected full plasmid sequences were deposited in a public database[80].

Other plasmids used for transfections in this work, mCh-FRB-miniTurbo, GRASP1-mCh and pLyn-FRB-mCh-nmd3-BSD (Addgene #85796) were generated by previous studies[32,73]. Used as templates or backbones for further cloning reactions, pSLI-N-GFP-2xFKBP-loxP (Addgene #85792), pSLI-C-GFP-Sandwich (also pSLI-sandwich, Addgene #85790), pSLI-C-mNG-Sandwich (same as addgene #85792 with mNG replacing GFP), pSkipFlox (Addgene #85797), mSc-Rab6_NLS-FRB, Sf3A2_mCh-Kelch13, pLyn-FRB-mCh-nmd3-BSD (Addgene #85796), were previously generated in our laboratory[32,33], pCAG_smFP-V5 was a gift from Loren Looger (Addgene #59758), and GAPM1-GFP and GAP45-mCh[60] were gifts from T. Gilberger (CSSB, Hamburg, DE).

## Selection linked integration (SLI)

The SLI approach was also used for endogenously editing all genes analyzed in this study, with slightly different plasmids depending on whether the modification was N- or C-terminal, as the N-terminal modifications require the insertion of a second recodonized version of the gene (as shown in Supplementary Fig. 1b) but the C-terminal ones do not. Methodologically, SLI was carried out as described[32]. Briefly, parasites were transfected with a SLI plasmid and selected with WR99210. When the parasites had resurfaced and were growing at usual rates, the drug in the medium was replaced by a different drug to which only parasites that had properly integrated the plasmid in their genome will be resistant. These drugs were G418 (in the case of C-terminal gene modifications) or DSM1 (in the case of N-terminal gene modifications). Medium and drug were changed daily for 5 days and cultures were periodically monitored, thereafter every second day until parasites resurfaced. Once integrated parasites were growing at usual rates, correct integration was evaluated by a diagnostic PCR from genomic DNA (collected using Monarch #T3010L kit). Diagnostic PCR involved reactions spanning the two integration junctions and the absence of the unmodified gene locus and were performed using Firepol enzyme (Solis biodyne #01-01-02000). Genomic sequences of the original and modified loci, including the primers and amplicon sizes for diagnostic PCR, are provided in Supplementary Data 6.

## Labeling of parasites and live-cell microscopy

For direct staining, parasites were incubated with 50 ng/ml Hoechst 33342 (Biomol #ABD-17533), 1 µM Tubulin Tracker Deep Red (Invitrogen #T34076) or 50 nM Halo-JF585 or -JFX650 ligands[81] (kind gift from L. Lavis, Janelia Farm, Ashburn VA) for nuclear, tubulin or Halo-

tag visualization, respectively. All staining reactions were carried out for 20 min at 37 °C in RPMI-1640 medium after which the parasites were washed once with medium, with an additional 10 min incubation and wash in the case of Halo dyes.

For wide-field fluorescence microscopy, parasites in RPMI medium were placed between a glass slide and a cover slip and imaged in a Zeiss AxioImager microscope equipped with a Hamamatsu Orca C4742-95 camera and a plan apochromat objective (63x, 1.4 NA, Oil DIC). Images were acquired using the AxioVision software (version 4.7) and processed using FIJI[82] to adjust for brightness and contrast and in a few cases apply a gaussian filter.

For laser scanning confocal microscopy of live cells was performed at 37 °C using an Olympus FluoView FV3000 system equipped with a universal plan apochromat objective (60×, 1.5 NA, oil) and a cell Vivo incubation system. Time lapse imaging of IMC formation was done as described[83]. Briefly, infected RBCs (34-38 hpi) were plated in a glass-bottomed dish (Ibidi #80427) previously coated with 0.5 mg/ml concanavalin A (Sigma-Aldrich #C0412) and allowed to attach for 10 min. Unbound cells were washed off with pre-warmed DPBS, the dish filled up completely with fresh medium (previously pre-adsorbed with uninfected RBCs), sealed with parafilm and placed into the microscope. Z-stacks were acquired every 20 minutes. Resulting images were processed with FIJI[82] or Imaris (Oxford Instruments), to adjust for brightness and contrast and apply a Gaussian filter.

## Ultrastructure expansion microscopy (U-ExM)

U-ExM was performed as previously described[62]. Parasites were placed on 3 mm PDL-coated coverslips and allowed to attach for 20 min at 37 °C. Samples were then washed with PBS, fixed with 4% formaldehyde in PBS for 20 min, and left overnight in post-fix solution [1.4% formaldehyde and 2% acrylamide (Merck #A4058)], all at 37 °C. The coverslips were then washed twice with PBS and placed cells facing down on 35 µl of monomer solution [19% sodium acrylate (Sigma-Aldrich #408220)}, 10% acrylamide and 0.1% N,N'-methylenebisacry-lamide (Merck #M1533)] with freshly added TEMED (EMD Millipore #1.10732) and APS (Sigma-Aldrich #A3678) to a final concentration of 0.5% each. Gels were allowed to polymerize at 37 °C for 1 h and were then incubated shaking at RT on denaturation buffer (200 mM SDS, 200 mM NaCl, 50 mM Tris-HCl pH 9). After the gels had separated from the coverslip, they were further incubated in denaturation buffer for 90 minutes at 95 °C, and were either froze at −20 °C or placed in water for three rounds of expansion of 30 min each. Expanded gels were shrunk by placing them in PBS for two 15-min rounds, and a slice of gel was cut for antibody staining. This slice was blocked with 3% BSA (Biomol #9048-46-8) in PBS for 30 min at RT, and left incubating with primary antibodies [anti-V5 (1/250, BioRad #MCA1360, clone SV5-Pk1, lot 162949), anti-αTubulin (1/500, Thermo Fischer #32-3500, clone B-5-1-2, lot WJ337893), anti-mCh (1/1000, abcam #ab167453, clone EPR20579, lot 1061130-4), anti-MORN1 (1/1000, a kind gift from B. Striepen[84])] shaking overnight at RT. Gels were then washed three times with 0.5% Tween in PBS, shaking for 10 minutes at RT, and incubated with secondary antibodies [anti-mouse IgG Alexa Fluor™ 594 (Thermo Fischer #A11032, lot 2301112), anti-mouse IgG Alexa Fluor™ 633 (Thermo Fischer #A21053, lot 948495), anti-rabbit IgG Alexa Fluor™ 546 (Thermo Fischer #A10040, lot 948483) and anti-rabbit IgG Alexa Fluor™ 647 (Thermo Fischer #A21244, lot 1386544), all used 1/500] and/or NHS-ester (1/250, Thermo Fischer #46403), Bodipy TR Ceramide (1/500, Invitrogen #B34400) and Hoechst 33342 for 3h at RT. Gels were washed three times with 0.5% Tween in PBS and were then placed on a glass-bottom dish (Ibidi #80427) previously coated with PDL for imaging in an Olympus FluoView FV3000 laser scanning confocal microscope equipped with a universal plan super apoc-hromat objective (60x, 1.3 NA, silicone). Images were processed using FIJI[82] to adjust brightness and contrast, and obtain Z-stack max intensity projections of the number of Z-slices specified in each image.

## DiQ-BioID

**Preparation of streptavidin sepharose beads for protease resistance.** Streptavidin sepharose beads were treated to be protease resistant for reducing the presence of streptavidin in the eluted peptides after on-bead trypsin digestion. This treatment was carried out as previously described by others[85]. Briefly, sepharose beads (Cytiva #17511301) were treated with cyclohexanedione (Merck #C101400) for 4 h at RT, washed with PBS-0.1% Tween (PBS-T), treated with a solution of 4% formaldehyde and 0.1 M of sodium cyanoborohydride (Sigma-Aldrich #8.18053) for 2 h at RT and washed with 0.1 M Tris-HCl pH 7.5 and twice with PBS-T.

**DiQ-BioID sample collection.** DiQ-BioID experiments were perfomed as previously outlined[33]. Briefly, cultures of genetically edited parasite episomally expressing the miniTurbo biotinyliser were grown to 200 mL in biotin-free RPMI-1640 medium (Biozol #USB-R9002-01). Asynchronous cultures were used for PfVAP and cultures synchronized to 34-38 hpi (by subsequent Percoll-Sorbitol synchronization to obtain 0-4 hpi rings) were used for PfVPS13L1. The culture was split into two, and in one half the dimerization of the biotinilyser with the POI (PfVPS13L1 or PfVAP) was induced by addition of 250 nM rapalog. Immediately after, 50 µM of biotin (Sigma-Aldrich #B4639) was added to both cultures for 30 minutes of labeling at standard growth conditions. Thereafter, RBCs were harvested, washed with 1xPBS and parasites were isolated by lysis with 0.03% saponin (in 1xPBS) on ice for 10 min, washed five times with 1xPBS and lysed in 2 ml of lysis buffer (50mM Tris-HCL pH 7.5, 500 mM NaCl, 1% Triton-X-100, 0.4% SDS) supplemented with 1 mM DTT, 1 mM PMSF and 1 x protease inhibitor cocktail (Roche #11836170001). The lysate was frozen at −80 °C until affinity purification was carried out.

**Affinity purification of biotin-labeled proteins.** Lysates were thawed and frozen twice, then centrifuged at $16,000 \times g$ for 10 to 60 min to clear the lysate. The supernatant was diluted 3-fold in 50 mM Tris-HCl pH 7.5 and incubated with 50 µl of pre-treated (for protease-resistance) streptavidin sepharose beads. The cleared lysate was incubated with the sepharose beads rotating overnight at 4 °C. The beads were washed twice with lysis buffer, once with dH$_2$0, twice with Tris-HCl pH 7.5 and three times in 100 mM TEAB (Sigma-Aldrich #T7408) pH 8.5, before adding 50 µl of elution buffer (2 M urea and 10 mM DTT in 100 mM Tris-HCl pH 7.5) and incubation for 20 min with shaking at 1400rpm at RT. On-bead digestion was then performed, first by adding 5 µl iodoacetamide to a final concentration of 50 mM and incubation for 10 min shaking in the dark for protein alkylation, and then by adding 500 ng of Trypsin (Sigma-Aldrich #T0303) and shaking for 2 h, all at RT. The supernatant was collected and the beads were resuspended in 50 µl of elution buffer and incubated for 15 min. The supernatant was again collected and combined with the previous one and 200 ng of additional trypsin was added and left shaking at RT overnight. Tryptic peptides were frozen at −80 °C and shipped to the EMBL proteomics core facility (Heidelberg, DE), where MS analysis was carried out.

**Preparation of peptides for MS analysis.** Peptides were dried and reconstituted in 10 µl of 100 mM Heps/NaOH, pH 8.5 and reacted for 1 h at room temperature with 80 µg of TMT6plex (Thermo Fischer #90066) dissolved in 4 µl of acetonitrile. Excess TMT reagent was quenched by the addition of 4 µl of an aqueous 5% hydroxylamine solution (Sigma-Aldrich #438227). Peptides were reconstituted in 0.1% formic acid, mixed and purified by a reverse phase clean-up step (OASIS HLB 96-well µElution Plate, Waters #186001828BA). Peptides were subjected to an offline fractionation under high pH conditions[86]. The resulting 6 fractions were analyzed on a Lumos system (Thermo Fischer).

**LC-MS/MS analysis.** Peptides were separated using an Ultimate 3000 nano RSLC system (Dionex) equipped with a trapping cartridge (Precolumn C18 PepMap100, 5 mm, 300 µm i.d., 5 µm, 100 Å) and an analytical column (Acclaim PepMap 100. 75 × 50 cm C18, 3 mm, 100 Å) connected to a nanospray-Flex ion source. The peptides were loaded onto the trap column at 30 µl per min using solvent A (0.1% formic acid) and eluted using a gradient from 2 to 38% Solvent B (0.1% formic acid in acetonitrile) over 52 min and then to 80% at 0.3 µl per min (all solvents were of LC-MS grade). The Orbitrap Fusion Lumos was operated in positive ion mode with a spray voltage of 2.4 kV and capillary temperature of 275 °C. Full scan MS spectra with a mass range of 375–1500 m/z were acquired in profile mode using a resolution of 60,000 with a maximum injection time of 50 ms, AGC operated in standard mode and a RF lens setting of 30%.

Fragmentation was triggered for 3 s cycle time for peptide-like features with charge states of 2–7 on the MS scan (data-dependent acquisition). Precursors were isolated using the quadrupole with a window of 0.7 m/z and fragmented with a normalized collision energy of 36%. Fragment mass spectra were acquired in profile mode and a resolution of 30,000 in profile mode. Maximum injection time was set to 94 ms or an AGC target of 200%. The dynamic exclusion was set to 60 s. Acquired data were analyzed using FragPipe[87] and a Uniprot *Plasmodium falciparum* isolate 3D7 fasta database (UP000001450, ID 36329, 5.372 entries, date: 06.03.2023, downloaded: 17.05.2023) including common contaminants. The following modifications were considered: Carbamidomethyl (C, fixed), TMT6plex (K, fixed), Acetyl (N-term, variable), Oxidation (M, variable) and TMT6plex (N-term, variable). The mass error tolerance for full scan MS spectra was set to 10 ppm and for MS/MS spectra to 0.02 Da. A maximum of 2 missed cleavages were allowed. A minimum of 2 unique peptides with a peptide length of at least seven amino acids and a false discovery rate below 0.01 were required on the peptide and protein level[88].

The raw output files of FragPipe were processed using the R programming language (ISBN 3-900051-07-0). Contaminants and reverse proteins were filtered out and only proteins that were quantified with at least 2 razor peptides were considered for the analysis. Log2 transformed raw TMT reporter ion intensities were first cleaned for batch effects using the 'removeBatchEffect' function of the limma package[89] and further normalized using the 'normalizeVSN' function of the limma package (VSN - variance stabilization normalization[90]). Proteins were tested for differential expression using a moderated t-test by applying the limma package ('lmFit' and 'eBayes' functions). The replicate information was added as a factor in the design matrix given as an argument to the 'lmFit' function of limma.

## Loss-of-function induction by knock sideways or conditional gene excision

Knock-sideways (KS) was induced by addition of 250 nM Rapalog to a cell line expressing the POI endogenously fused to 4 FKBP domains and an episomally expressed FRB protein anchored to the PM via a Lyn-targeting region. Mislocalization to the PM was assessed by microscopy after 5 hours of induction. In the case of PfVPS13L1, as the IMC and the PM are in close proximity, the mislocalization was assessed by comparison with the IMC marker PhIL1 (no overlap: full mislocalization; overlap only of 1-2 PfVPS13L1 puncta: partial mislocalization).

Gene excision was induced by addition of 250 nM Rapalog to a cell line expressing PfVAP flanked by two loxP sites at its endogenous locus (Supplementary Fig. 1b–c) and an episomally expressed diCre protein using pSkipFlox as done previously[32]. Excision was assessed by PCR one cycle after induction (Supplementary Fig. 1g), using primers annealing to sites outside the loxP flanked region which would result in a reduced amplicon size upon correct gene excision (Supplementary Data 6).

## Parasite growth assays

Importance of proteins for parasite survival was tested by monitoring the growth of asynchronous control and diCre-excised or KS parasite cultures by flow cytometry-based measurements of parasitemia every day over 7 (for diCre excision) or 5 (for KS) days. Samples were stained with Hoechst 33342 and Dihydroethidium (Cayman #12013-5) for 20 min at room temperature, and the parasites were inactivated and the staining stopped with 400 µl of 0.003% glutaraldehyde (Roth #4157) containing medium. Parasitemia was measured as the percentage of positively stained cells over 100000 events counted with an LSRII flow cytometer (BD Biosciences) with a FACSDiva software (BD Biosciences) as described[32,91].

For stage specific growth assays, parasites were first synchronized. For GFP-2xFKBP-PfVAP[endo], ring stage parasites were synchronized to 10-18hpi by two 10-minute incubation with 5% sorbitol 10 hours apart. For PfVPS13L1-GFP-SW[endo], ring stage parasites were synchronized to 0-4hpi by an initial Percoll enrichment of schizonts and sorbitol four hours after adding the parasites to fresh erythrocytes[83]. KS or diCre excision was induced with 250 nM Rapalog (Takara Bio), and Giemsa (Merck) smears were taken at the indicated time points.

## Egress and invasion assays

PfVPS13L1-GFP-SW knocked-in parasites were synchronized by Percoll followed by sorbitol after four hours. Parasites were grown for another 34 h (34-38hpi), at which point they were separated in two cultures, one which was treated with Rapalog and one kept as a control. Six hours after (40-44hpi), 2 µM Compound 2[66,92,93] (imidazopyridine referred to as Compound 2 (4-[7-[(dimethylamino)methyl]-2-(4-fluorophenyl)imidazo [1,2-α]pyridine-3-yl]pyrimidin-2-amine), a kind gift from Michael J. Blackman [MRC-NIMR, London, UK]) was added to arrest parasites at the segmenter stage. Giemsa smears were prepared at 44-48hpi (pre-egress), the Compound 2 was washed off, parasites were allowed to egress and invade new erythrocytes for 4 h, and a second set of Giemsa smears was prepared at 48-52hpi (post-egress). The number of rings and schizonts in both sets of smears was counted and used to calculate the number of new rings in the post-egress sample per schizont in the pre-egress sample.

## FFAT prediction and AlphaFold3 mediated filtering

Filtering of PfVAP DiQ-BioID hits using FFAT motif identification was done by sequence scoring and AlphaFold3[41] predicted binding of the motif to PfVAP. First, the lowest-scoring motif (i.e., most likely to be binding partner) was identified for each of the protein hits. Prediction of FFAT motifs within a protein sequence was carried out using the scoring matrix previously proposed based on motifs identified in yeast and human proteins[40]. A Python code written in collaboration with ChatGPT (OpenAI, 2023) was used and is deposited in a public repository[80]. Briefly, the code scores any sequence of 7 amino acid residues following the scoring matrix, giving a value to each position of the motif based on the consensus sequence. The negative charge of the 6 residues upstream of the potential motif is also considered. Motifs scoring 4 or below are considered potential FFAT binders and were considered for AlphaFold3-mediated filtering. For this, structural predictions of the MSP domain and the selected FFAT motif interactions were obtained using the AlphaFold3 server, and motifs with an interface predicted template modeling (ipTM) of 0.7 were selected for further analysis. Full-length PfVAP and the full-length protein candidates containing the selected motifs were then submitted to the AlphaFold3 server and the structural predictions with an ipTM value of 0.45 or above (corresponding to the higher half of all the ipTM values) were selected as likely PfVAP binders. The 13 motifs remaining after this filtering were selected for episomal expression in Halo-PfVAP[endo] parasites to assess binding.

## Sequence and structural analysis

Protein sequences were retrieved from PlasmoDB (release 67). For conservation analysis across *Plasmodium* species (FFAT-containing proteins in Fig. 3d and PfVPS13L proteins in Supplementary Fig. 6a), sequences from reference strains of *P. falciparum*, *P. berghei*, *P. knowlesi*, *P. ovale*, *P.malariae*, *P. vivax-like*, *P. vivax*, *P. gallinaceum*, *P. chabaudi*, *P. yoelii* and *P. coatneyi* were considered. Maximum-likelihood phylogenetic tree of PfVPS13L proteins was generated using MEGA11[94] with 100 bootstrapping repeats.

LTPs were identified by sequence and structure homology, using HHPred[95] and Foldseek[96]. Structures were downloaded from the AlphaFold public database[97,98], for the shuttle-like LTPs, or predicted using AlphaFold3[41], for the BLTPs, and analysed using PyMOL[99]. For the predictions, the BLTP sequences were split in 2 or 3 overlapping segments, which were then aligned in PyMOL to get the full-length structure of PfVPS13L1. The structures obtained were deposited in a public repository[80]. Shuttle-like LTP structures were aligned with indicated structures in Supplementary Fig. 4a using TM-align[100]. For the interaction predictions, the PH domain of PfVPS13L1 and the HEPN-L domain of PfAegerolysin were input in AlphaFold3[41]. The results of the prediction and the alignment used for the conservation-based colored representations were deposited in the Zenodo public database[80].

## Statistics and reproducibility

For qualitative microscopy experiments, at least 10 images per sample/replicate were collected across 3 replicates for epifluorescence microscopy, and across 2 replicates for confocal and ultrastructure expansion microscopy. For microscopy images used for quantifications, at least 20 images were acquired across 3 replicates, with the exception of the images for the PH binding to the PPM in Fig. 6h and Supplementary Fig. 10i, in which cases only 10 and 14 images across 3 replicates were considered for the ones co-expressing the HEPN-L domain, and only 5 images across 1 replicate were considered for the ones expressing the PH domain alone, as repeated qualitatative analysis using epifluorescence microscopy had already determined a cytosolic signal so no further replicates were added after confirmation of such a signal in randomly chosen cells. In most experiments, all images were considered for the analysis, with the exception of the IMC perimeter measurements which were taken from 15–19 cells per sample/replicate chosen randomly from the total of cells.

## Reporting summary

Further information on research design is available in the Nature Portfolio Reporting Summary linked to this article.

## Data availability

The *Plasmodium falciparum* protein database used in this study can be accessed at PlasmoDB[101], release 67. The mass spectrometry proteomics data have been deposited to the ProteomeXchange Consortium via the PRIDE[102] partner repository with the dataset identifier PXD066159. AlphaFold3[41] predicted structures have been deposited in the Zenodo public repository[80]. All other data are included in the article and Supplementary Data files. Source data are provided. Source data are provided with this paper.

## Code availability

Python code for FFAT analysis has been deposited in the Zenodo public repository[80].

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

## Acknowledgements

We thank J. S. Wichers-Misterek for essential discussion before the beginning of this project. We thank J. Cubillán-Marín and I. Henshall, from the malaria cell biology laboratory (BNITM, Hamburg) and T. Gilberger (CSSB, Hamburg) for discussion and G. Farias, T. Gilberger (CSSB, Hamburg) and B. Striepen (University of Pennsylvania, PA) for sharing of reagents. A. Guillén-Samander was supported by an EMBO long-term postdoctoral fellowship (ALTF 166-2022).

## Author contributions

A.G.S. and T.S. conceptualized the project. A.G.S. designed and performed most experiments and analysed the data. N.P. performed the FFAT filtering; V.H., H.M.B. and J.P.M.R. each generated one parasite line used in this study; H.O.R.C. identified the interaction between PfVPS13L1 and PfAegerolysin; A.R.H. contributed to data analysis; P.H. and F.S. performed the LC-MS/MS analysis. A.G.S. prepared the figures. The original draft was written by A.G.S., revised and edited by T.S., and reviewed by all the authors.

## Funding

## Competing interests

The authors declare no competing interests.
