## [Peer Review file · Nature Communications]

A bridge-like lipid transfer protein is critical for generation of invasive stages in malaria parasites

Corresponding Author: Dr Tobias Spielmann

Version 0:

Reviewer comments:

Reviewer #1

(Remarks to the Author)

The manuscript by Guillén-Samander and colleagues investigates the function of a putative lipid transfer protein, VPS13L1. The authors searched for proteins that may be involved in lipid transfers at membrane contact sites within the ER of *Plasmodium falciparum* using PfVAP as the starting point. The study builds upon the observation in model eukaryotic cells that lipids are synthesized in the ER and transported to other regions of the cell by vesicles or lipid transfer proteins. With PfVAP proximity labeling, the authors identified a novel bridge-like lipid transfer protein (VPS13L1). The bulk of the manuscript focuses on the characterization of this protein. The authors use several complementary approaches to convincingly demonstrate the protein interaction and localization of multiple candidates. VPS13L1 is essential for asexual replication, shown by knocksideways system.

The findings are quite exciting and provide some real novel insights into how the inner membrane complex in *Plasmodium falciparum* is formed. The results are important and will be of interest to both parasitologists and cell biologists.

Major points:

1. Line 154-158 makes an intriguing statement about using AlphaFold3 to predict structures of large unknown proteins to identify LTPs. This is a remarkable use of the predicted protein structure to predict function. I think it would be helpful to provide some more details for what aspects of the predicted fold structure was used to identify the protein as a putative bridge-like LTP. I may be misunderstanding the claims. If this is the case, then it should be explained more clearly.
2. It would be useful to provide some additional colocalization markers to more clearly identify the contact site or contact region with the IMC. I suggest that the authors use an epitope-tagged basal complex protein, like BTP1 or CINCH, that would help to demonstrate that the contact point is the basal side of the IMC. The model proposed in 6 would be better supported with this additional data.

Minor points:

1. PfVPS13L1 is a terrible name for a protein.
2. Fig 1g "Percentage of PM circumference at contact with ER" would be better to say "Percentage of PM circumference in contact with ER".
3. The setup of fig 2h is a little confusing. Perhaps the labels LynGFP-FFAT and Halo-VAP could be horizontal above the two rows of images. As it is currently presented, it looks like the top row is the GFP and the bottom row is the Halo. Obviously, this is not the case, but it could be presented more clearly.
4. In figure 5k, the GAP45-mCh staining is unfortunately not very clean, at least in the max-projections that are shown. It is not a major issue because the results are very clear. However, if possible, it would be nice to improve these images. This could be done by using a different epitope tag besides mCherry or by using a primary antisera against GAP45 or other IMC protein.

Reviewer #2

(Remarks to the Author)

Overall the experiments and data are very well done, but a general issue that affects different parts of the work is that the way it is written makes it hard to penetrate for anyone who is not an expert in *Plasmodium*. Explanations of normal cell

morphology and development (similar to Fig 6) need to appear at the start and be better/fully labelled.

Points about the experiments/results:

Fig 2f:

Is there a shift in FFAT score distribution as the group of proteins is narrowed from 115>29>13>9 ? Show this for the Slee matrix and possibly for the new matrix (see comments below)

Fig 2i

“difference to consensus opisthokont motif in positions 1 and -1.” The way this is described, in particular: repeating sequences in 2i already present in the central rows of 2g and making a sequence logo from them, implies that the authors believe Plasmodium has its own flavour of the FFAT-like motif. While there is little evidence against that idea, there is also not much evidence for it. This is the first time a species has been suggested to show such variation so as to necessitate a species-specific scoring system.

a. Position -1: negatively charged residue in position -1

I cannot see how this differs materially from the established consensus? (See Di Mattia et al., 2020 PMID: 33124732)

b. Position +1: a lack of importance of the amino acid here

The finding is that position 1 has an anionic residue (DE) in only 2 of 7 motifs, while a hydrophobic residue (ILY) is present in the other 5 of the 7 motifs. The 7 motifs are the subgroup that test positive for VAP binding in cells, while 4 other motifs test negative from a subgroup of 11, which were derived from the much larger group of 185 physical interactors of VAP where a FFAT-like motif and the whole protein both could be modelled by AF3 to bind VAP reasonably well.

To my mind this falls a long way short of showing that motifs in Plasmodium differs from other species

If this was meaningful, I would expect a difference in Pf VAP compared to human/yeast/A. thaliana. Such a difference in VAP would correlate structurally with the change in the motifs. Does Pf VAP have a different charge distribution or a unique set of substations in the key strands that make up the most conserved 16 residues of VAP, the so called ‘VAP conserved sequence’? In my searches I find nothing like this. Instead, the VAP conserved sequence was as follows

VAPA Hs FKVKTTAPRRYCVRPN

align FK+KTTAP Y VRP+

Pf FKIKTTAPNNYLVRPS

All 3 residues with non-conservative changes also vary in human MOSPD2, which binds the same targets as VAPA and VAPB (Di Mattia et al., 2018 PMID: 29858488).

Another way to consider if D/E are important in position 1 is to look at conservation for the single motif that the authors have tested in context of the whole protein. Vps13 in other alveolates does not have hydrophobic residues at position 1:

ETSSSA-SFVSAED (F=888) in A0A9P1GTI7 Cladocodium goreau

CLSDDL-DFFDACS (F=897) in U6GU52 Eimeria acervulina

While it is interesting that 5 of the 7 positives among the 185 screened had hydrophobic at position 1, and also that the enrichment of motifs in the set of 185 increased with an adjusted scoring motif, the claim here should be supported by data showing that D/E at position 1 fails to strengthen the interaction: does substitution of D1577 in Vps13L1 with I/L reduce affinity for Pf VAP? It would be worth doing the same experiment with human/yeast VAP in case the lack of importance of D/E at residue is a previously overlooked general principle, despite previous papers such as Furuita et al., 2010 PMID:20178991 showing that this residue interacts with the MSP domain. Interestingly their E1A substitution only reduced affinity two-fold. Elsewhere, Mikitova and Levine (2012) reported that changing A1 in a plant motif (AFFDTRD) to M1 made little difference, while H1 and especially K1 reduced affinity.

All these results are consistent with those obtained here, and inconsistent with the new proposed scoring matrix, but might warrant a different path: changing position 1 in the universally used scoring matrix to more accurately reflect the whole literature!

Fig 3a/c:

PF3D7_1351000 (PfPITP) has a FFAT-like motif: DYNESED (Y167) in the middle of a loop not participating in the PITP domain structure: 156-197. This scores similarly or better than the one in PfOSBP, so should be included in some way.

Pf VAP hits in Suppl File1/3

Give a few more details how the other 2 VPS13L proteins, those with no identifiable sequence similarity, were found. This will be helpful to guide others to do something similar in other cases and help us judge that they are genuine hits.

L 127: to help readers, say here what proteins these 2 identifiers refer to

Fig 3d:

I could not locate domain maps of the other 2 Vps13L proteins L3 and L4. Also, where AF3 is uninformative, say if HHpred finds even a weak alignment or a confident secondary structural element. Somewhere in the MS say how many amino acids the 2 VPS13L proteins have that are missing from this figure.

Fig S6: part E (and elsewhere): show both channels in B&W, not just one, and then a colour merge

Fig 6 bottom half, diagrams in centre and right: what do the long curved arrows indicate?

Discussion

say more about where the results might be improved to test hypotheses by further experiments:

Does acute knock-sideways or destruction of Vps13L1 cause dissociation of IMC from ER?

Does expression of Vps13L1 with specific mutations prevent rescue of knock-sideways: mutations could include hydrophilic bands across the hydrophobic groove.

Signed:

Tim Levine

Version 1:

Reviewer comments:

Reviewer #1

(Remarks to the Author)

The authors have provided excellent responses to my previous major suggestions. I appreciate the effort to provide additional explanation of their AlphaFold3 protein searches. I also appreciate the generation of multiple transgenic parasites to respond to previous requests. I have only minor, mostly style comments now:

1. Line 302, 310, 311, etc. -- BTP1/MORN1 are more commonly described as a marker of the basal complex (BC) and not the basal body (BB). The BB is organizing center for axonemes or flagella.
2. Line 304, 601 -- the technique is Ultrastructure expansion microscopy, not "Ultra-expansion microscopy"

Reviewer #2

(Remarks to the Author)

Generally the Authors have done a very good job addressing my comments.

In discussing the effect of the new mutations at positions 1 and -1, the new sentence "This apparent flexibility might be considered during FFAT motif scoring." (Line 161) undersells the current work. Better would be something more definitive like "This apparent importance of position -1 above that of position +1 might be considered during a future reanalysis of how to score FFAT motifs." This would require inclusion of the data in the figure in the rebuttal - I see no reason to exclude it.

"PfVPS13L2 also had a high scoring FFAT motif" - I could not find the coordinates. If it is centred around Y602 (DYFTTKE) then bear in mind that T604 (and other nearby serines) might only be phosphorylated in some conditions.

One point from the other reviewer.

Name PfVPS13L1 - Here I agree with the Authors

REVIEWER COMMENTS

Reviewer #1 (Remarks to the Author):

The manuscript by Guillén-Samander and colleagues investigates the function of a putative lipid transfer protein, VPS13L1. The authors searched for proteins that may be involved in lipid transfers at membrane contact sites within the ER of *Plasmodium falciparum* using PfVAP as the starting point. The study builds upon the observation in model eukaryotic cells that lipids are synthesized in the ER and transported to other regions of the cell by vesicles or lipid transfer proteins. With PfVAP proximity labeling, the authors identified a novel bridge-like lipid transfer protein (VPS13L1). The bulk of the manuscript focuses on the characterization of this protein. The authors use several complementary approaches to convincingly demonstrate the protein interaction and localization of multiple candidates. VPS13L1 is essential for asexual replication, shown by knocksideways system.

The findings are quite exciting and provide some real novel insights into how the inner membrane complex in *Plasmodium falciparum* is formed. The results are important and will be of interest to both parasitologists and cell biologists.

We thank the reviewer for the time invested in reviewing the manuscript and the constructive comments on our work.

Major points:

1. Line 154-158 makes an intriguing statement about using AlphaFold3 to predict structures of large unknown proteins to identify LTPs. This is a remarkable use of the predicted protein structure to predict function. I think it would be helpful to provide some more details for what aspects of the predicted fold structure was used to identify the protein as a putative bridge-like LTP. I may be misunderstanding the claims. If this is the case, then it should be explained more clearly.

We now clarified how these BLTPs were found and give detailed information about the structural elements found and what was used to make the call that these proteins are VPS13-like proteins (new Extended Data Fig. 4). Structurally, VPS13-like proteins have a long rod with an internal hydrophobic groove that is formed by a repeating unit of 5 beta-strands and 1 alpha helix (RBG repeat), and further C-terminal adapter domains, namely a VAB domain formed by 6 repeats of a beta-strand-formed unit, an ATG2-C, formed by amphipathic helices, and a PH domain (PMID: 36571082). Despite the conserved structure in the *Plasmodium* proteins, sequence similarity with known opisthokont VPS13 proteins is scarce. The most conserved region is the one known as Chorein, composed by the first two RBG repeats. Using an alignment of sequences of this region and HHpred on all predicted *P. falciparum* proteins, we identified four VPS13 proteins, PfVPS13L1-4. Two of these, PfVPS13L1 and L2 were part of the PfVAP DiQ-BioID (HHPred similarities in these are now highlighted in the Extended Data Fig. 4a).

The two bridges in the PfVAP DiQ-BioID that had no apparent sequence homology were found using the AF3 approach the reviewer refers to. Initially, we attempted to do a structural based search using algorithms like Foldseek, however the VPS13 proteins tend to be large and proteins above 2700aa are not included in the AlphaFold database, preventing the use of these approaches. Hence, we individually predicted the structures of all large unknown proteins in the *Plasmodium* genome using the AlphaFold3 server. Two of these, in the manuscript named VPS13L5 and 6, contained folded regions with similarities to RBG repeats and the VAB domain. However, *P. falciparum* sequences contain many long asparagine-rich repeats, which are predicted as unfolded and hamper the prediction of the RBG repeats as a continuous rod. These repeats, however, are not conserved in other *Plasmodium* species. Hence, we predicted the structure of the homologues of VPS13L6 in *P. gallinaceum* and found a more clearly formed RBG rod which supports that these *P. falciparum* proteins are VPS13-like.

We have adjusted the text accordingly to explain the identification of the VPS13 like proteins more clearly, and in addition we have included snapshots of the structures of the four *P. falciparum* and one *P. gallinaceum* VPS13L proteins to illustrate the approach and show the identified elements in the new Extended Data Fig. 4.

2. It would be useful to provide some additional colocalization markers to more clearly identify the contact site or contact region with the IMC. I suggest that the authors use an epitope-tagged basal complex protein, like BTP1 or CINCH, that would help to demonstrate that the contact point is the basal side of the IMC. The model proposed in 6 would be better supported with this additional data.

We thank the reviewer for this comment and we agree that determining if the contact point is in the basal body of the IMC would be of interest. As per the suggestion of the reviewer we generated new cell lines to address this. We episomally expressed BTP1-mNG in the parasite line expressing PfVPS13L1 endogenously modified to carry a Halo tag. While there is a colocalization in very early stages of IMC formation, PfVPS13L1 does not seem to migrate with the basal body (this result is now included as Fig. 4g). To determine where the “contact point” in the IMC is, we attempted to localize PfVPS13L1 by U-ExM. For this we generated a parasite line with the highly sensitive smV5 tag on its C-terminus and performed co-staining with an anti-MORN1 antibody (basal body marker). Some punctae of the PfVPS13L1 signal were observed in proximity to the basal body but not precisely colocalizing, as expected for a general IMC localization. This is now included as Fig. 4h.

Since we have now identified a direct interaction between the C-terminus of PfVPS13L1 and PfAegerolysin (Fig. 5f-h), and we know from our fluorescence microscopy experiments that the expression of the latter is higher than that of PfVPS13L1 (albeit still quite low), we attempted to use PfAegerolysin to gain insight into the potential contact point of the PfVPS13L1 bridge at the IMC. We generated a line with PfAegerolysin tagged with a smV5 tag on its endogenous locus. This protein was detected by U-ExM throughout the IMC and not necessarily colocalizing with the basal body, and with a more obvious localization in early stages of IMC formation. This data confirms the co-localization results of PfVPS13L1 with basal body markers and is now included in Fig. 5c-d. Taken together, our data points towards the ER-IMC MCS not being at a known IMC subsection, but rather happening at multiple contact points throughout the IMC.

Minor points:

1. PfVPS13L1 is a terrible name for a protein.

We agree that the name is not very appealing nor does it reflect the specific function presented here. However, despite this being the first study for this protein in protozoa, yeast and human VPS13 proteins have been studied for several years (Park JS, et al. PMID: 22442115, Lang AB, et al. PMID: 26370498, Kumar N, et al. PMID: 30093493) and, as the founding members of the superfamily of

bridge-like LTPs, are currently a topic of great interest in membrane cell biology. Hence, giving these proteins a different name in the parasite would disrupt the connection with this body of work. While BLTP (short for bridge-like lipid transfer protein) was recently proposed as new nomenclature for the human VPS13 genes, the HUGO Gene Nomenclature Committee approved name remained unchanged to maintain the connection with the previous body of work (Braschi B, et al. PMID: 36461115).

2. Fig 1g "Percentage of PM circumference at contact with ER" would be better to say "Percentage of PM circumference in contact with ER".

We thank the reviewer for the suggestion and have corrected the figure.

3. The setup of fig 2h is a little confusing. Perhaps the labels LynGFP-FFAT and Halo-VAP could be horizontal above the two rows of images. As it is currently presented, it looks like the top row is the GFP and the bottom row is the Halo. Obviously, this is not the case, but it could be presented more clearly.

We thank the reviewer for the suggestion and have corrected the figure.

4. In figure 5k, the GAP45-mCh staining is unfortunately not very clean, at least in the max-projections that are shown. It is not a major issue because the results are very clear. However, if possible, it would be nice to improve these images. This could be done by using a different epitope tag besides mCherry or by using a primary antisera against GAP45 or other IMC protein.

We thank the reviewer for suggesting this. Indeed, the staining with the mCherry antibody is not very clean, especially in the "Control" segmenters when compared to the Rapalog treated, likely because the GAP45-mCh protein is distributed throughout a wider surface area in the more expanded IMC. We generated a line where GAP45 was tagged with an smV5 protein and the staining is much cleaner. However, as both the V5 and the Tubulin antibodies are from mouse, we cannot use this line for a staining of both structures in parallel. Since we think the current image in Figure 5k is more informative on the loss-of-function phenotype as it shows the tubulin structures, we are keeping this panel in the main Figures and including the GAP45-smV5 staining in the Extended Data Fig. 10f.

Reviewer #2 (Remarks to the Author):

Overall the experiments and data are very well done, but a general issue that affects different parts of the work is that the way it is written makes it hard to penetrate for anyone who is not an expert in Plasmodium. Explanations of normal cell morphology and development (similar to Fig 6) need to appear at the start and be better/fully labelled.

We thank Dr. Levine for the time invested in reviewing our manuscript and the constructive comments on our work. We appreciate this suggestion and have now added an additional schematic of *P. falciparum* parasite development in RBCs, highlighting key organelles for this study (new Extended Data Fig 1a and also a schematic of IMC growth in Fig. 4i). While the part in Extended Data Fig 1a might be useful to show in figure 1 it would make this figure very crowded. This is the reason for placing it into the supplement.

Points about the experiments/results:

Fig 2f:

Is there a shift in FFAT score distribution as the group of proteins is narrowed from 115>29>13>9 ? Show this for the Slee matrix and possibly for the new matrix (see comments below)

This is an interesting point. Indeed, there is a shift in FFAT score distribution as the proteins are narrowed, especially when applying the AF3 motif iPTM score filter (i.e. hits filtered from 115 to 29). This would indicate that higher-scoring motifs are more likely to interact with PfVAP. However, the mean of the scores of the binders is relatively high, as many of the binder motifs have a score of 3, suggesting that any motif scoring 3 or below should be considered for PfVAP interaction analysis. This, together with more details on the filtering steps, are now included in the Extended Data Figure 2k-l. The shift is observed for both matrices, but since the new matrix is now excluded from the manuscript (see details below), the FFAT score distribution for this matrix is only included in this response. With the new matrix, the shift is more pronounced for the binding motifs, reflecting the bias towards the confirmed binding motifs.

Fig 2i

"difference to consensus opisthokont motif in positions 1 and -1." The way this is described, in particular: repeating sequences in 2i already present in the central rows of 2g and making a sequence logo from them, implies that the authors believe Plasmodium has its own flavour of the FFAT-like motif. While there is little evidence against that idea, there is also not much evidence for it. This is the first time a species has been suggested to show such variation so as to necessitate a species-specific scoring system.

We agree with the reviewer that we do not present enough evidence for a Plasmodium-specific FFAT-like flavor, as our analysis is only based on 7 identified motifs in *P. falciparum*. We had done this as we observed that the binding motifs scored quite high (i.e. the mean and median score of binding motifs is 2.4 and 3, respectively), and considered that this could reflect a small change in affinity of the binders. Further explanation about why we decided to remove the new proposed matrix is included below.

a. Position -1: negatively charged residue in position -1

I cannot see how this differs materially from the established consensus? (See Di Mattia et al., 2020 PMID: 33124732)

We were of the impression that the weight given to position -1 in the established consensus was not so high and that there were also binding motifs with non-negatively charged amino acids in that position in Di Mattia et al., 2020. As all of our binding motifs in *P. falciparum* had a negatively charged residue in that position, we interpreted this as a potential need for a negatively charged residue and therefore we proposed to give a bigger weight to this position in the new matrix for this

parasite. However, as stated above, we understand that this new matrix is biased towards the 7 identified binding motifs and we have removed it from the revised version of the manuscript.

b. Position +1: a lack of importance of the amino acid here

The finding is that position 1 has an anionic residue (DE) in only 2 of 7 motifs, while a hydrophobic residue (ILY) is present in the other 5 of the 7 motifs. The 7 motifs are the subgroup that test positive for VAP binding in cells, while 4 other motifs test negative from a subgroup of 11, which were derived from the much larger group of 185 physical interactors of VAP where a FFAT-like motif and the whole protein both could be modelled by AF3 to bind VAP reasonably well.

To my mind this falls a long way short of showing that motifs in Plasmodium differs from other species

If this was meaningful, I would expect a difference in Pf VAP compared to human/yeast/A. thaliana. Such a difference in VAP would correlate structurally with the change in the motifs. Does Pf VAP have a different charge distribution or a unique set of substations in the key strands that make up the most conserved 16 residues of VAP, the so called 'VAP conserved sequence'? In my searches I find nothing like this. Instead, the VAP conserved sequence was as follows

```
VAPA Hs  FKVKTTAPRRYCVRPN
align   FK+KTTAP  Y VRP+
Pf      FKIKTTAPNNYLVRPS
```

All 3 residues with non-conservative changes also vary in human MOSPD2, which binds the same targets as VAPA and VAPB (Di Mattia et al., 2018 PMID: 29858488).

Another way to consider if D/E are important in position 1 is to look at conservation for the single motif that the authors have tested in context of the whole protein. Vps13 in other alveolates does not have hydrophobic residues at position 1:

```
ETSSSA-SFVSAED (F=888) in A0A9P1GTI7 Cladocopium goreau
CLSDDL-DFFDAC (F=897) in U6GU52 Eimeria acervulina
```

While it is interesting that 5 of the 7 positives among the 185 screened had hydrophobic at position 1, and also that the enrichment of motifs in the set of 185 increased with an adjusted scoring motif, the claim here should be supported by data showing that D/E at position 1 fails to strengthen the interaction: does substitution of D1577 in Vps13L1 with I/L reduce affinity for Pf VAP? It would be worth doing the same experiment with human/yeast VAP in case the lack of importance of D/E at residue is a previously overlooked general principle, despite previous papers such as Furuita et al., 2010 PMID:20178991 showing that this residue interacts with the MSP domain. Interestingly their E1A substitution only reduced affinity two-fold. Elsewhere, Mikitova and Levine (2012) reported that changing A1 in a plant motif (AFFDTRD) to M1 made little difference, while H1 and especially K1 reduced affinity.

All these results are consistent with those obtained here, and inconsistent with the new proposed scoring matrix, but might warrant a different path: changing position 1 in the universally used scoring matrix to more accurately reflect the whole literature!

We thank the reviewer for these comments and suggestions. During the revision process we attempted to test how changes in individual positions affected the binding of the motif to PfVAP (specifically tested using the PfVPS13L1 motif). We generated lines to test mutations in position 1 (from D to K and D to V) and position -1 (from D to K). All of the tested motifs seemed to recruit PfVAP to the PM (see graph included below), and the one where position -1 was switched to a K showed differential binding between experimental repeats. This is the only motif with which we have observed different binding across experiments, which could indicate a lower affinity or an unexpected selection across time of parasites where the motif was not bound. Unfortunately, since our assay relies on episomal overexpression of the motif, it is likely not suitable for testing small differences in affinity, but rather only concluding binding vs no binding. The reason for this is that a slightly higher expression of a lower binding affinity motif might yield similar results in terms of ER recruitment than a low-expressed but

high binding affinity motif. Taking this into account, and considering the low likelihood of certain species requiring a genus-specific scoring matrix as pointed out by the reviewer, we conclude that we cannot claim differences in affinity without being able to experimentally test such proposed differences. Hence, we have removed the *Plasmodium*-centered scoring matrix from our manuscript. We agree that the finding of an hydrophobic residue in position 1 in many motifs is still of interest and we have rewritten the results text as follows:

“Alignment of the interacting motifs and their orthologous sequences in 10 different Plasmodium species revealed a slight difference from the consensus motif⁴⁰ (Fig. 2i): the amino acid in position 1, proposed to favor a negatively charged residue, was taken by a hydrophobic residue in 5 out of the 7 motifs identified in Plasmodium proteins. This apparent flexibility might be considered during FFAT motif scoring.”

Fig 3a/c:

PF3D7_1351000 (PfPITP) has a FFAT-like motif: DYNESED (Y167) in the middle of a loop not participating in the PITP domain structure: 156-197. This scores similarly or better than the one in PfOSBP, so should be included in some way.

We thank the reviewer for pointing this out. As we focused only in the motifs that passed the AlphaFold3 based filters, we had overlooked FFAT motifs with high sequence score within the identified LTPs. We noticed that, besides the one in PfPITP mentioned by the reviewer, there is also a high-scoring motif in PfVPS13L2 (PF3D7_1343800). Both of these motifs were tested for PfVAP binding but showed no binding. These results have been included in Extended Data Fig. 3d.

Pf VAP hits in Suppl File1/3

Give a few more details how the other 2 VPS13L proteins, those with no identifiable sequence similarity, were found. This will be helpful to guide others to do something similar in other cases and help us judge that they are genuine hits.

We have modified the text and added a new figure (Extended Data Fig. 4). For details please see response to a similar point raised by reviewer 1 (Major comment #1 from Reviewer 1).

L 127: to help readers, say here what proteins these 2 identifiers refer to
We thank the reviewer for the suggestion and have added the names of the proteins the identifiers refer to.

Fig 3d:

I could not locate domain maps of the other 2 Vps13L proteins L3 and L4. Also, where AF3 is uninformative, say if HHpred finds even a weak alignment or a confident secondary structural element. Somewhere in the MS say how many amino acids the 2 VPS13L proteins have that are missing from this figure.

We included only the domain maps of the lipid transfer proteins found in the PfVAP BioID. VPS13L3 and L4 were not identified in the BioID (although by mistake the previous version of the article mentioned L3 in the plot, and this has been fixed now). We consider that including the other two domain maps might lead readers into thinking that these proteins were identified in proximity to PfVAP. For this reason we did not include them. However, we have now added the size of the proteins next to their accession code in the phylogenetic tree of Extended Data Fig. 5. Additionally, we have indicated the regions where HHpred gave a result for the identified PfVPS13L1 and 2 proteins (HHpred failed to identify any similarity in L5 and L6) in Extended Data Fig. 4a.

Fig S6: part E (and elsewhere): show both channels in B&W, not just one, and then a colour merge
We thank the reviewer for the suggestion and have corrected the figure.

Fig 6 bottom half, diagrams in centre and right: what do the long curved arrows indicate?
These arrows indicate trafficking from the Golgi. We have now indicated this in the legend.

Discussion

say more about where the results might be improved to test hypotheses by further experiments:
Does acute knock-sideways or destruction of Vps13L1 cause dissociation of IMC from ER?
Does expression of Vps13L1 with specific mutations prevent rescue of knock-sideways: mutations could include hydrophilic bands across the hydrophobic groove.

The limitations of our study and open questions are now discussed in a new paragraph which reads as follows:

“Although our phenotypic results are compatible with PfVPS13L1 fueling IMC growth through lipid transfer ER-IMC MCSs, some questions remain. Given the confined space in the parasite, particularly in schizonts, direct contact between the ER and IMC is difficult to determine and it is for instance unclear whether the contact is broken upon loss of PfVPS13L1. In addition, while knock-sideways allows for acute loss-of-function, it also involves the relocation of the BLTP to the PM, where a gain-of-function defect cannot be fully discarded. While episomal expression to complement the knock sideways would be an option to address this, doing this for such a large protein is challenging. Hence, the development of a genetic-based model for phenotypic rescue would be advantageous and could also open the possibility of studying the effect of specific mutations, such as e.g. in the interaction interfaces or on the hydrophilic bands across the RBG groove which produce a gain of function in yeast.”

Signed:
Tim Levine

REVIEWERS' COMMENTS

We appreciate both reviewers' constructive comments throughout this review process as they have significantly improved our manuscript.

Reviewer #1 (Remarks to the Author):

The authors have provided excellent responses to my previous major suggestions. I appreciate the effort to provide additional explanation of their AlphaFold3 protein searches. I also appreciate the generation of multiple transgenic parasites to respond to previous requests. I have only minor, mostly style comments now:

1. Line 302, 310, 311, etc. -- BTP1/MORN1 are more commonly described as a marker of the basal complex (BC) and not the basal body (BB). The BB is organizing center for axonemes or flagella.
2. Line 304, 601 -- the technique is Ultrastructure expansion microscopy, not "Ultra-expansion microscopy"

We thank the reviewer for their comments on our work and we have corrected the text for both points accordingly.

Reviewer #2 (Remarks to the Author):

Generally the Authors have done a very good job addressing my comments.

In discussing the effect of the new mutations at positions 1 and -1, the new sentence "This apparent flexibility might be considered during FFAT motif scoring." (Line 161) undersells the current work. Better would be something more definitive like "This apparent importance of position -1 above that of position +1 might be considered during a future reanalysis of how to score FFAT motifs." This would require inclusion of the data in the figure in the rebuttal - I see no reason to exclude it.

We thank the reviewer for their comments.

We had opted to leave the data regarding mutations in positions -1 and 1 out as we are unsure whether the motif with the mutation in position -1 binds or not, as the inconsistency between experiments could reflect a limitation of the method in detecting small differences in affinity. We have now included the data as new Supplementary Figure 3 and modified the text. In the future, a more careful examination using a different method could be of interest to define whether there are any differences in FFAT binding affinities when compared to known plant and opisthokont VAP orthologues.

"PfvPS13L2 also had a high scoring FFAT motif" - I could not find the coordinates. If it is centred around Y602 (DYFTTKE) then bear in mind that T604 (and other nearby serines) might only be phosphorylated in some conditions.

The FFAT motif identified in PfvPS13L2 is the one centered around Y603. All the motifs are available in the Supplementary Data 2 and 3 files, but we have also now included coordinates and core sequence of the ones tested in the legend for Supplementary Figure 3d. Indeed, it would be interesting to test whether this motif (and others) could bind PfvVAP in response to phosphorylation.

One point from the other reviewer.

Name PfvPS13L1 - Here I agree with the Authors